# Untwisting the *Caenorhabditis elegans* embryo

Ryan Patrick Christensen[1][*][†], Alexandra Bokinsky[2][†], Anthony Santella[3][†], Yicong Wu[1][†], Javier Marquina-Solis[4], Min Guo[1,5], Ismar Kovacevic[3], Abhishek Kumar[1,4], Peter W Winter[1], Nicole Tashakkori[1], Evan McCreedy[2], Huafeng Liu[5], Matthew McAuliffe[2], William Mohler[6], Daniel A Colón-Ramos[4], Zhirong Bao[3], Hari Shroff[1]

[1]Section on High Resolution Optical Imaging, National Institute of Biomedical Imaging and Bioengineering, National Institutes of Health, Bethesda, United States; [2]Biomedical Imaging Research Services Section, Center for Information Technology, National Institutes of Health, Bethesda, United States; [3]Developmental Biology Program, Sloan-Kettering Institute, New York, United States; [4]Program in Cellular Neuroscience, Neurodegeneration and Repair, Department of Cell Biology, Yale University School of Medicine, New Haven, United States; [5]State Key Laboratory of Modern Optical Instrumentation, College of Optical Science and Engineering, Zhejiang University, Hangzhou, China; [6]Department of Genetics and Developmental Biology, University of Connecticut Health Center, Farmington, United States

**Abstract** The nematode *Caenorhabditis elegans* possesses a simple embryonic nervous system with few enough neurons that the growth of each cell could be followed to provide a systems-level view of development. However, studies of single cell development have largely been conducted in fixed or pre-twitching live embryos, because of technical difficulties associated with embryo movement in late embryogenesis. We present open-source untwisting and annotation software (http://mipav.cit.nih.gov/plugin_jws/mipav_worm_plugin.php) that allows the investigation of neurodevelopmental events in late embryogenesis and apply it to track the 3D positions of seam cell nuclei, neurons, and neurites in multiple elongating embryos. We also provide a tutorial describing how to use the software (*Supplementary file 1*) and a detailed description of the untwisting algorithm (Appendix). The detailed positional information we obtained enabled us to develop a composite model showing movement of these cells and neurites in an 'average' worm embryo. The untwisting and cell tracking capabilities of our method provide a foundation on which to catalog *C. elegans* neurodevelopment, allowing interrogation of developmental events in previously inaccessible periods of embryogenesis.

**\*For correspondence:** ryan.christensen@nih.gov

[†]These authors contributed equally to this work

**Competing interests:** The authors declare that no competing interests exist.

## Introduction

Understanding how complex neural circuits and entire nervous systems form is one of the fundamental goals of neuroscience. While substantial progress has been made in identifying guidance factors in neurodevelopment (*Kolodkin and Tessier-Lavigne, 2011*; *Dudanova and Klein, 2013*; *Chilton, 2006*; *O'Donnell et al., 2009*), how known factors interact to direct the formation of complex neural circuits remains mysterious (*Dudanova and Klein, 2013*). Examining the entirety of neurodevelopment in intact, living samples would be useful in understanding larger scale principles that orchestrate nervous system formation. Unfortunately, technological limitations and inherent nervous

**eLife digest** Understanding how the brain and nervous system develops from a few cells into complex, interconnected networks is a key goal for neuroscientists. Although researchers have identified many of the genes involved in this process, how these work together to form an entire brain remains unknown.

A simple worm called *Caenorhabiditis elegans* is commonly used to study brain development because it has only about 300 neurons, simplifying the study of its nervous system. The worms are easy to grow in the laboratory and are transparent, allowing scientists to observe how living worms develop using a microscope. Researchers have learned a great deal about the initial growth of the nervous system in *C. elegans* embryos. However, it has been difficult to study the embryos once their muscles have formed because they constantly twist, fold, and move, making it hard to track the cells.

Now, Christensen, Bokinsky, Santella, Wu et al. have developed a computer program that allows scientists to virtually untwist the embryos and follow the development of the nervous system from its beginning to when the embryo hatches. First, images are taken of worm embryos that produce fluorescent proteins marking certain body parts. The program, with user input, labels the fluorescent cells in the images, which indicates how the embryo is bending and allows the program to straighten the worm. The program can also track how cells move around the embryo during development and show the positional relationships between different cells at different stages of development.

Christensen et al. have made the program freely available for other researchers to use. The next step is to increase automation, making the software faster and more straightforward for users. Ultimately, the software could help in the challenge to comprehensively examine the development of each neuron in the worm.

system complexity have hindered our ability to capture a 'systems-level' view of the developing brain.

One model organism well-suited to systems-level neuroscience research is *Caenorhabditis elegans*, which possesses a simple nervous system comprising 302 neurons (*White et al., 1986*), 222 of which form during embryogenesis (*Sulston et al., 1983*). The adult connectome has been reconstructed, and the morphology of all adult neurons has been mapped at electron-microscopy resolution (*White et al., 1986*); the genome sequenced (*C. elegans* Sequence Consortium, 1998); and the organism is genetically tractable and transparent at all life stages, enabling investigation with light microscopy. The simplicity of the *C. elegans* nervous system, its experimental accessibility, and the extensive knowledge base make it a promising candidate for following the development of all neurons in the embryo, and eventually understanding associated molecular mechanisms. The resulting 'neurodevelopmental atlas' would represent the first view of how an entire nervous system forms.

Despite the potential of the nematode as a model, imaging neurodevelopment (*Wu et al., 2013a*) throughout embryogenesis is challenging due to embryonic sensitivity to photodamage and photobleaching, limiting imaging to several hours on most systems; the need for subcellular spatial resolution due to the small size of the embryo; and motion blur caused by rapid embryo movement after muscular twitching begins. Once images are captured, data analysis poses new problems: while it would be easy to assemble an atlas of neuronal positions and morphology if all cells were easily identifiable in one animal, techniques that allow imaging with single-cell contrast (such as Brainbow [*Livet et al., 2007*]) are unavailable in the nematode. Currently, any attempt to build a neurodevelopmental atlas would require imaging small numbers of non-overlapping, easily distinguishable neurons, and finding methods to combine the data from multiple embryos into a composite whole. To our knowledge, comprehensive solutions to these problems do not yet exist.

Recent advances in light-sheet fluorescence microscopy (LSFM [*Santi, 2011*]) have solved many of the imaging problems outlined above. LSFM sweeps a thin sheet of light through the sample, relying on perpendicular detection of fluorescence. This geometry allows far more rapid imaging and reduced phototoxicity relative to confocal microscopy (*Huisken et al., 2004*; *Holekamp et al.,*

*2008*), enabling the use of LSFM in a variety of transformative applications. These include recording whole-brain calcium signaling in larval zebrafish (*Ahrens et al., 2013*; *Ito, 2013*), and imaging (*Wu et al., 2011*; *Keller et al., 2008*) and tracking (*Amat et al., 2014*; *Bao, 2006*; *Santella et al., 2010*) large numbers of cells in developing embryos. Multiple LSFM implementations now obviate the problems of motion blur and photo damage in worm embryos (*Wu et al., 2011*; *Wu et al., 2013*; *Kumar et al., 2014*; *2015*), and also offer sufficient spatiotemporal resolution (sub-μm in all three spatial dimensions, sub-second volumetric imaging [*Wu et al., 2013*; *Kumar et al., 2014*]) that subcellular morphology may be observed over the entire 14-hour period of embryogenesis. Despite these advances, morphological changes still pose problems when trying to follow individual cells, or when combining data from multiple embryos.

To address these problems, we have generated a nematode strain that expresses fluorescent markers within specific cells, and designed software that uses these markers to computationally 'untwist' the embryo, resulting in straightened volumes that significantly ease the tracking of developmental events in later embryonic stages (described briefly in a preliminary conference proceeding [*Christensen, 2015*]). Our open-source software is based on the NIH's Medical Image Processing, Analysis, and Visualization (MIPAV [*McAuliffe, 2001*; *Haak et al., 2015*]) platform, implemented as a standalone plugin (http://mipav.cit.nih.gov/plugin_jws/mipav_worm_plugin.php). Computational untwisting algorithms have previously been used to straighten images of L1 larval worms for use in tracking nuclear position (*Peng et al., 2008*; *Long et al., 2009*) in both two and three dimensions, but to our knowledge, these algorithms are not suitable for the nematode embryo. In addition to the untwisting capability, our plugin includes the ability to annotate and track 3D positions over time, allowing semi-automated quantification of cell and neurite positions in twisted (and untwisted) embryos. The positional data so derived also facilitate comparison and combination of information from multiple embryos, allowing us to create a composite model of development.

We demonstrate the capabilities of our method by computationally untwisting eight nematode embryos; tracking the position of seam cell nuclei, the canal-associated neuron (CAN), ALA, and AIY neuron cell bodies, and the growing neurites of the ALA neuron in the untwisted reference frame; and combining the data from multiple embryos to model the time-evolution of all these elements within the elongating embryo. We find that seam cell nuclear positions are highly stereotyped across different embryos, while the rate of elongation varies according to position along the embryo. Of the neurons we examined, ALA and AIY move in concert with neighboring seam cell nuclei, suggesting they are passively 'dragged' with the rest of the elongating worm embryo, while the CAN neurons actively migrate at a faster rate than the surrounding seam cell nuclei. Tracking ALA neurites reveals that anterior-posterior neurite outgrowth starts toward the end of elongation and continues after cells reach their final positions. Our method is the first to track cell positions in the context of the entire embryo, from the beginning of twitching until hatching. We anticipate that our software will significantly further the ability to examine *C. elegans* development in the post-twitching regime and lay a foundation for understanding the formation of the *C. elegans* nervous system.

## Results

In order to computationally straighten an embryo, an essential first step is defining limits of the growing worm body, thus specifying how the embryo folds inside the eggshell. Nematode embryos undergo both bending and helical twisting around the nose-to-tail axis (*Figure 1—figure supplement 1*) posing challenges in untwisting the embryo relative to larval or adult nematodes. Our approach uses fluorescent markers driven by cell-specific promoters to define the boundaries of the worm body. We use a seam cell marker (*SCM::GFP*) to label the 20–22 seam cell nuclei, identifying the left and right sides of the worm; and a *dlg-1::GFP* fusion protein to label apical gut junctions and hypodermal junctions, revealing the locations of the anterior tip of the pharynx (hereafter referred to as the nose), tail, midline, and hypodermal cell boundaries (*Figure 1A,B*). This combination of markers allows automated segmentation of seam cells and manual identification of the nose, tail, and sides of the worm, thus enabling us to model the twisted, bent embryo within the eggshell, and serving as a basis for computationally untwisting the worm (*Figure 1C*).

We used a dual-view selective plane illumination microscopy (diSPIM) implementation of LSFM to capture images of developing embryos (*Wu et al., 2013*; *Kumar et al., 2014*). The diSPIM was

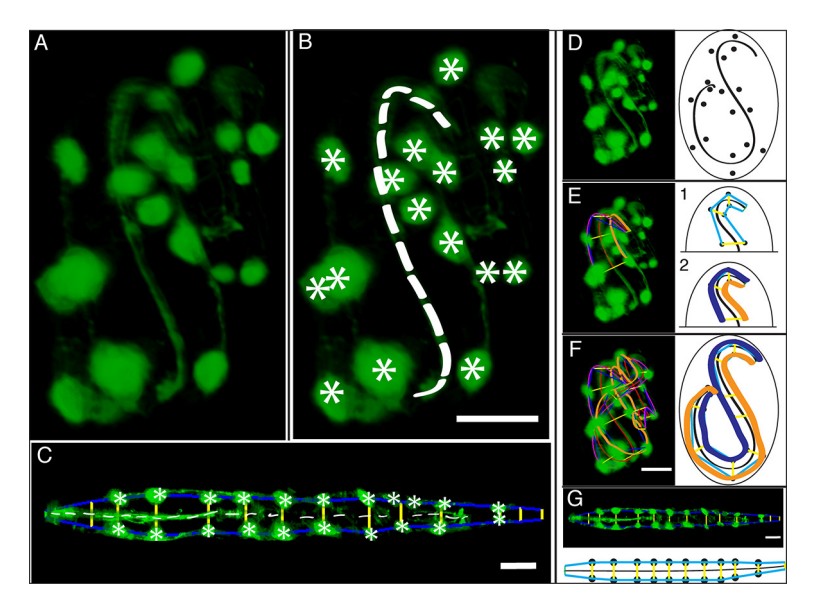

**Figure 1.** Key steps in worm untwisting. (**A**) An image of a threefold embryo in the twisted state, showing the untwisting markers. (**B**) The same image as in (**A**) with the untwisting markers labeled. Asterisks mark seam cell nuclei, and the dashed line indicates the midline marker. (**C**) The same embryo as in (**A, B**), after untwisting. Asterisks and dashed line as in **B**. (**D–F**) Further detail lattice creation and splines that model embryo. (**D**) Left: same embryo volume as in (**A**). Right: accompanying schematic showing the seam cell nuclei in the twisted embryo (black circles) and midline (interior black line). (**E**) Lattice creation. As diagrammed in right schematic, parts (1) and (2), the user adds points to create a lattice (blue and yellow lines). After the lattice is built, the algorithm generates splines defining the edges of the worm (orange and purple lines) automatically. The midline is also defined with a spline (red line at left). (**F**) The embryo volume and accompanying schematic showing a completed lattice and model. (**G**) The embryo volume and accompanying schematic after untwisting. All scale bars: 10 µm.

The following figure supplements are available for figure 1:

**Figure supplement 1.** Helical twisting in the nematode embryo.

**Figure supplement 2.** DiSPIM is useful in identifying landmarks in the twisted embryo.

**Figure supplement 3.** Effects of lattice point number on untwisting results.

**Figure supplement 4.** Untwisting a larval nematode.

chosen due to the combination of high-imaging speed and isotropic resolution that it provides, making the identification of cells and cellular structures in a twisted-up embryo significantly easier than with lower resolution alternatives (such as single-view light-sheet microscopy, *Figure 1—figure supplement 2*). After images are acquired in the diSPIM, registered, and deconvolved, a user begins untwisting by downloading and running our software (http://mipav.cit.nih.gov/plugin_jws/mipav_worm_plugin.php, *Supplementary file 1*).

First, seam cell nuclei are automatically detected, segmented, and paired to create candidate lattices. Seam cell segmentation and lattice-building are manually verified by a user, who can also incorporate additional information derived from pharyngeal and hypodermal markers, which are difficult to automatically segment (*Figure 1D,E*). Several possible lattices are generated, and the five most likely to be correct are displayed to the user for selection and editing of the correct lattice. The resulting lattice is used to generate a 3D model of the worm volume (*Figure 1F*, *Video 1*). In cases where automated lattice-building fails, lattices can be built manually by marking the positions of seam cell nuclei, nose, bends in the embryo, and tail. When manually building lattices, minimally 22 +2B lattice points are recommended (22 is the number of lattice points corresponding to seam cell

nuclei, plus a pair of points to mark the nose, and B is the number of bends between seam cell nuclei in the embryo). Fewer lattice points than the number of seam cell nuclei gives unphysical, short volumes, and more than ~32 points does not noticeably improve quality in the untwisted volumes (*Figure 1—figure supplement 3*).

The first step in creating the 3D model is to generate curves defining the center and sides of the worm. The centerline curve is uniformly sampled to generate a series of planes extending along and normal to the curve, while avoiding overlap within the model. This series of restricted planes comprises the worm model and is updated as new lattice points are added. To generate a straightened volume, the voxels in the original image that intersect with the sampling planes in the worm model are captured, and the sampling planes and associated voxels are concatenated in the head-to-tail direction to generate a straightened volume (*Figure 1G*, *Video 1*). The same process can be used to straighten images of older animals (such as L2 larvae, *Figure 1— figure supplement 4*). More details are provided in *Supplementary file 1* and Appendix.

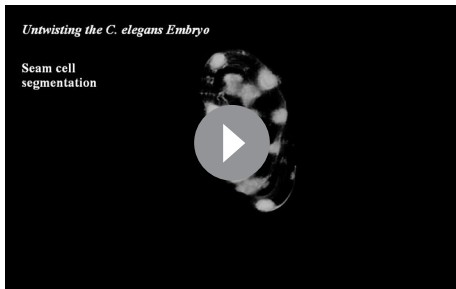

**Video 1.** Sequential steps used in the automated lattice-building plugin. This animation provides a graphical representation of the computational steps used to segment seam cells, build a lattice, and straighten embryo volumes. For additional information refer to *Supplementary file 1* and Appendix.

In addition to untwisting, it is also useful to obtain the 3D position of a cell or point of interest within the nematode embryo. Thus, our software also includes an annotation capability, allowing a user to define points within the embryo for which they would like to obtain 3D coordinates both before and after untwisting (*Supplementary file 1*). The user adds annotations similarly to lattice points, marking the volume location where the desired point should appear. The user must also add an origin point from which the relative 3D position of all other points is calculated from. As pharyngeal labeling is consistent and bright in most diSPIM volumes, we use the nose as a standard origin in all datasets described in the paper. Once the origin and annotation points have been defined, the user can untwist the worm and obtain the 3D coordinates of each annotation point in a spreadsheet file.

In order to ensure that our algorithm did not alter the distance between portions of the embryo during the untwisting process, we compared the apparent 3D distance between, or along, landmark features within twisted and untwisted embryo volumes (embryos 1–6, *Figure 2*). First, we determined the distances between nuclei in seam cell pairs (*Figure 2A,B*). If untwisting did not effect morphology, we reasoned that these distance should be conserved regardless of whether the embryo is twisted or untwisted. We measured the difference between pair distance in twisted- and untwisted datasets at every fifth or tenth time point for both the first (H0) and last (T) pairs of seam cells in six different embryos, reasoning that the difference should be close to 0. The apparent untwisted distance between seam cell pairs H0 and T closely mirrored the values in the twisted worm, with the population average difference across timepoints and embryos ($<\mu_{\text{Difference, time}}>_{\text{embryo}} \pm$ population standard deviation $<\sigma_{\text{Difference, time}}>_{\text{embryo}}$) for H0 0.4 μm $\pm$ 0.3 μm, and for T 0.3 $\pm$ 0.2 μm (*Figure 2C*, *Figure 2—figure supplement 1*, *Supplementary file 5*, Materials and methods). The largest difference at any individual timepoint between twisted and untwisted values was 1.7 μm for H0 and 1.2 μm for T.

Since the model of the twisted embryo is based on positional coordinates of the seam cell nuclei, we would expect these paired distances in twisted- and untwisted embryos to agree. For a more stringent control, we also assessed the apparent distance between nose and the pharynx-gut transition (effectively the pharyngeal contour length) in twisted and untwisted embryos (*Figure 2A,B*). Although the pharynx is not used as a landmark for defining the worm model used in untwisting, we still expect its contour length to be conserved despite untwisting. Here, too, we measured a close correspondence (typically less than 5% of the total pharyngeal length). The population $<\mu_{\text{Difference, time}}>_{\text{embryo}} \pm <\sigma_{\text{Difference, time}}>_{\text{embryo}}$ between twisted and untwisted pharyngeal lengths was 2.5 μm $\pm$ 1.6 μm (with the maximum difference between the untwisted and twisted values for any individual

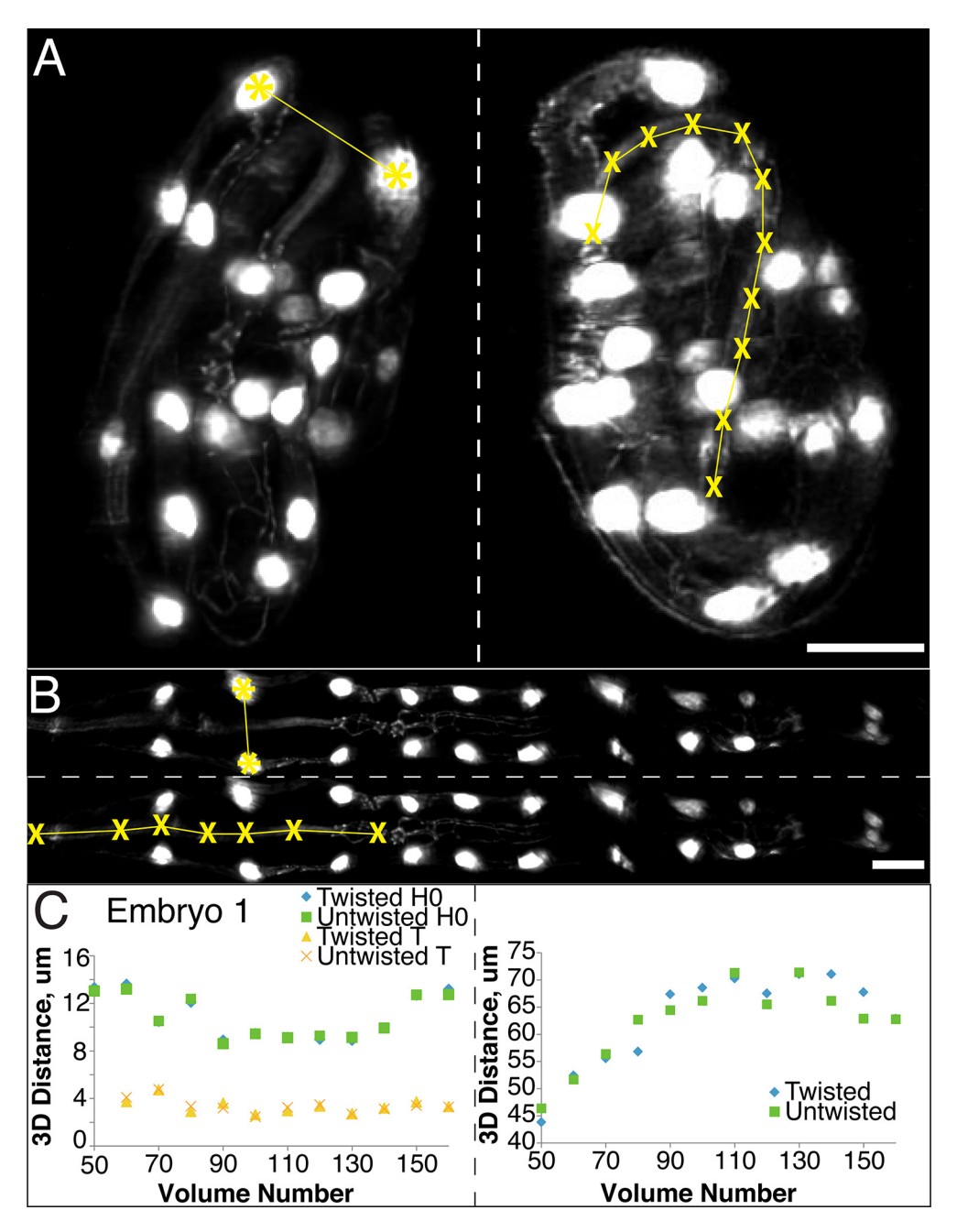

**Figure 2.** The untwisting algorithm accurately preserves embryo dimensions. Distances between seam cell nuclei (left) and pharyngeal lengths (right) were compared in twisted (**A**) and untwisted (**B**) worm embryos. All scalebars: 10 μm. (**C**) Comparative 3D distance measurements of seam cell nuclei pairs H0 and T (left graphs) and pharyngeal lengths (right graphs) for one representative embryo (a comparison across six different embryos is presented in *Figure 2—figure supplement 1*). In all cases, distance measurements in the twisted case are within 5 μm of distance measurements in the untwisted case.

The following figure supplement is available for figure 2:

**Figure supplement 1.** Untwisting does not systematically alter worm morphology

timepoint being 8.8 μm, *Figure 2C*, *Figure 2—figure supplement 1*). We conclude that our untwisting procedure accurately captures distances present in the twisted embryo.

The combination of untwisting and annotation capabilities we developed allows the analysis of overall morphological changes in a developing embryo and the precise tracking of positions for individual cells or subcellular structures. We first examined overall morphological changes in the nematode embryo. Embryos lengthened (from 86 ± 5 μm at early 1.5-fold, measured from the nose to the tail, to 162 ± 19 μm within the last 30 min before hatching, measured from the nose to the last pair of seam cells, mean ± standard deviation [SD], 5 embryos) and narrowed in width (measured diameter across the widest cross-section 22 ± 1 μm at early timepoints, and 16 ± 1 μm at late timepoints, mean ± SD, 5 embryos) as they progressed from comma stage to late-3 fold (*Figure 3A–H*, *Figure 3—figure supplement 1*; *Figure 3—figure supplement 2*; *Video 2*). We used our software to manually annotate and extract the positional trajectories of seam cell nuclei during this time period, as they moved relative to the nose of the animal (*Figure 3J–3L*, *Figure 3—figure supplement 1*). We note that seam cell V5 divides late in the threefold embryo into Q and V5 daughters; in such cases, we tracked the anteriormost daughter, Q, and thus refer to V5 as Q/V5 in our paper. The motion of seam cell nuclei followed relatively simple trends that were easily evident, despite the noise present in the raw untwisted trajectories. During elongation, seam cell nuclei moved laterally ('X' motion, *Figure 3J*) towards the midline, while maintaining a relatively fixed dorso-ventral position ('Y' motion, *Figure 3K*). Along the axial, head-to-tail axis, the displacement of seam cell nuclei was biphasic, showing a fast, approximately linear dependence on time, followed by a slower plateau ('Z' motion, *Figure 3L*) (*Priess and Hirsh, 1986*; *Chin-Sang and Chisholm, 2000*; *Ding et al., 2004*; *Norman and Moerman, 2002*). While embryo elongation has been examined before (*Priess and Hirsh, 1986*), our method is the first that allows 3D interrogation of whole, live, untwisted nematodes at arbitrary timepoints in embryogenesis (*Figure 3*, *Video 2*).

The strong qualitative similarities in seam cell nuclear trajectories among the five embryos we inspected led us to investigate whether data from different embryos could be combined to yield a composite model of development representing growth in an 'average' embryo. Initial examination of the axial (nose-to-tail) seam cell nuclear trajectories from different embryos suggested a high degree of stereotypy; except for a relative shift in time, the trajectories displayed very similar shapes (*Figure 4A*). We thus shifted the axial data in time until the trajectories from multiple embryos overlaid (*Figure 4B*, *Figure 4—figure supplement 1*). We determined the amount of shift by using a three parameter logistic function to fit the raw axial displacement data (*Figure 4—figure supplement 2*, *Tables 1*, *2*, Materials and methods), overlaying the data from various embryos until the inflection points in each curve were identical.

We applied the same time shift to the X- and Y- seam cell nuclear coordinates, finding that seam cell nuclear positions followed similar trajectories throughout elongation (average SD calculated across all 20 seam cell nuclei and all timepoints, $<<\sigma_X>_{time}>_{seam\ cell}$ 0.8 μm, $<<\sigma_Y>_{time}>_{seam\ cell}$ 0.7 μm, $<<\sigma_Z>_{time}>_{seam\ cell}$ 4.6 μm, see also *Figure 4—figure supplement 3*, *Table 1*, and 'Materials and methods'). After shifting, we averaged (*Figure 4C*) and fitted (*Figure 4D*, *Figure 4—figure supplement 4*, *Table 1*, *Supplementary file 2*) the embryo XYZ trajectories, thus generating positions representing the noise-free time evolution of seam cell nuclei. We note that the choice of fitting functions is somewhat arbitrary. For axial positions, the growth that we and others (*Priess and Hirsh, 1986*) have observed leads to a sigmoidal fitting function. Amongst the various three-parameter sigmoidal functions (*Table 2*), we found that the three-parameter logistic function gave the best qualitative and quantitative (*Figure 4—figure supplement 2*) agreement with the data. We fitted lateral ('X') seam cell nuclei positions with a two parameter power law function, and dorso-ventral ('Y') positions with a linear function, as empirically these functions described our data well. Despite the *ad hoc* nature of these fits, we found that fitted values were within 1.5 μm of the X, Y averaged data, and within 7.5 μm of the Z averaged data (*Supplementary file 3*). For reference, the total length of the untwisted embryo at the final time point was 162.0 ± 18.7 μm (mean ± SD, 5 embryos), measured from the nose to the last pair of seam cells, and the corresponding diameter at the last time point 16.1 ± 1.3 μm, measured at the widest cross-section in the animal.

The averaged, fitted seam cell nuclei data allowed us to inspect the relative relationships among seam cell nuclei in an elongating embryo (*Figure 5A*, *Videos 3,4*). Since we fixed the nose as the stationary origin in our untwisting procedure, this location does not move in 4D representations of the fitted embryo. In this 'nose-centric' reference frame, points further from the origin also appear

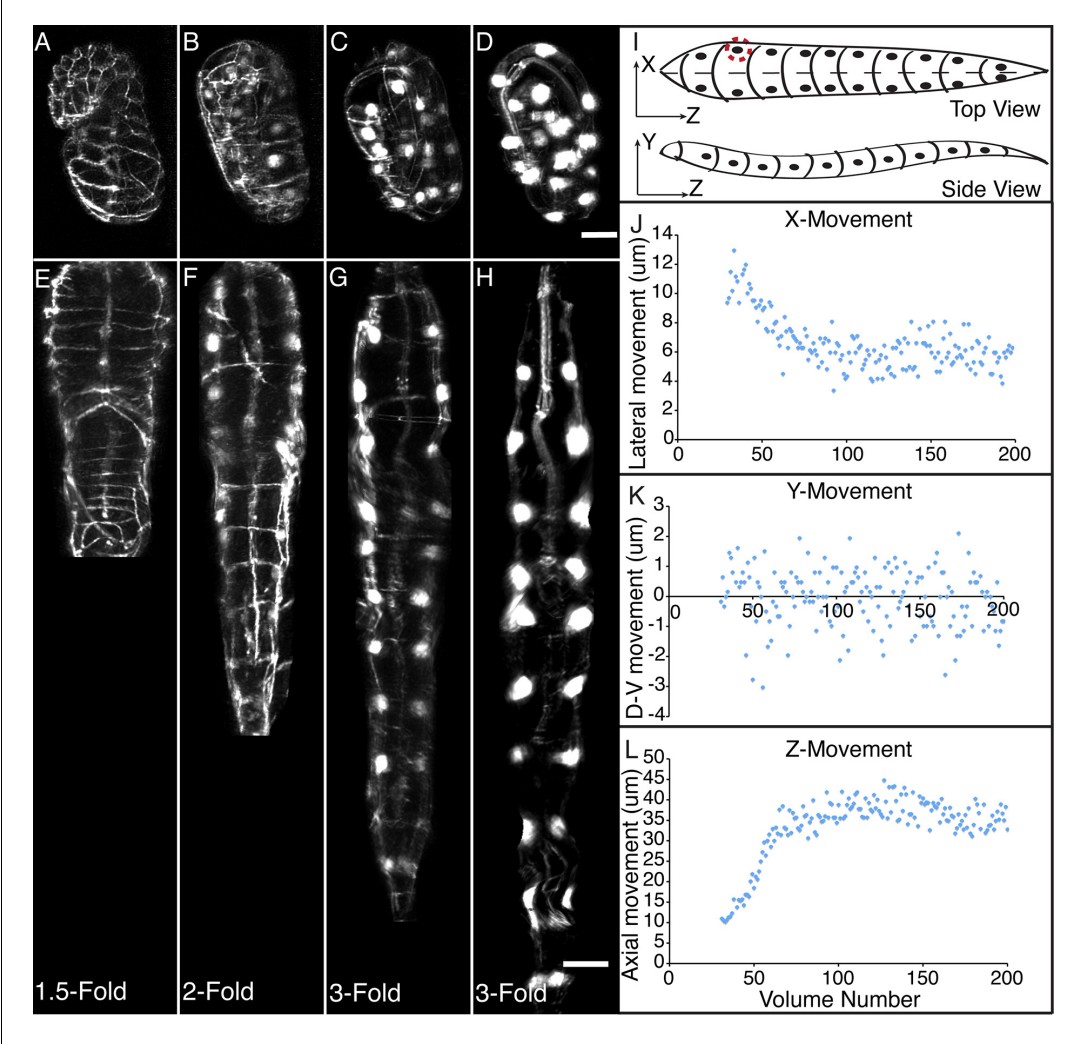

**Figure 3.** Morphological changes in embryonic development, as unveiled by untwisting algorithm. Selected volumetric timepoints pre (A–D) and post (E–H) untwisting, with canonical state of embryo indicated at bottom. See also *Video 2*. (I) Cartoon of untwisted embryo, indicating coordinate system. (J–L) X, Y, and Z movements of circled seam cell nucleus in (I). Measurements are indicated relative to the animal's nose, fixed as the origin in all untwisted datasets. All scalebars: 10 µm.

The following figure supplements are available for figure 3:

**Figure supplement 1.** Comparison of untwisted 1.5-fold embryos after shifting.

**Figure supplement 2.** Comparison of threefold embryos after shifting.

**Figure supplement 3.** Data Post-processing.

to move faster and farther than points closer to the origin. To better understand the growth rates of individual seam cell nuclei in relation to their neighbors, and the overall length changes within the elongating embryo in a frame-independent manner, we also computed the differences in position between adjacent pairs of nearest-neighbor seam cell nuclei over time (*Figure 5B–D*, *Figure 5—figure supplement 1*). In 'X' and 'Y' dimensions, seam cell nuclei exhibited similar movement patterns, remaining largely stationary in 'Y' (*Figure 5—figure supplement 1*), and moving inwards (toward

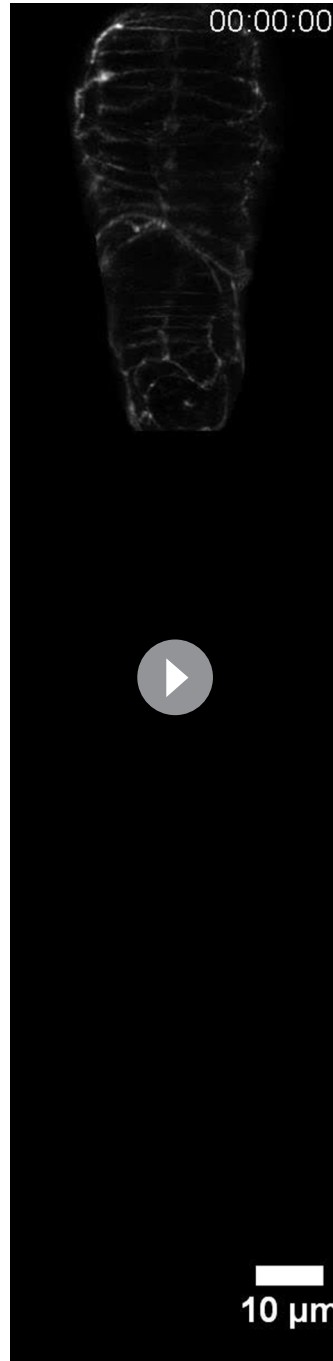

00:00:00

10 µm

**Video 2.** Raw data showing an untwisted worm developing from the 1.5-fold stage until hatching. Despite errors in individual untwisted volumes, the overall pattern of embryonic development and elongation is clear.

the origin) in 'X' (*Figure 5—figure supplement 1*) at similar rates. In contrast, seam cell nuclei movement along the 'Z' direction was more heterogeneous. For example, the distance between the origin and nuclei of seam cell pair H0, measured from the fitted data, changed from 2.4 µm to 23.8 µm over elongation (*Figure 5B*), while the distance between seam cell nuclear pairs V6 and T remained essentially constant, at 22.5 µm (*Figure 5D*). Thus, the rate of increase in distance between the origin and H0 was significantly greater than the increase in distance between V6 and T, over the same period. Other adjacent nuclear pairs separated at roughly similar rates from start to end of elongation (these pairs increased in distance 6.8 ± 2.8 µm, mean ± SD from 7 adjacent pairs of seam cell nuclei, again derived from the fitted data in *Figure 5C*). These trends were not the results of artifacts in our fitting procedure, as they were evident also in the raw, averaged data (compare left and right graphs in *Figure 5B-D*). The apparent differences in X- and Z- pre- and post-elongating seam cell nuclei positions that we observe are consistent with the asymmetric morphology of the pre-elongating embryo. Since the embryo starts out in a tadpole-like shape with the head larger than the tail, the seam cell nuclei in the head must move a greater distance than the nuclei in the tail to achieve a uniform diameter in the elongated embryo.

Embryo elongation is thought to be dependent on an actin-based contractile mechanism (*Priess and Hirsh, 1986*). The complex, position-dependent motion we observed is likely inconsistent with a simple, uniform contraction, as this phenomena cannot explain our finding that different regions of the embryo elongate at markedly different rates. To our knowledge, current models of embryo elongation have not taken into account the differential elongation we observed across the worm body. We expect that incorporating additional data derived from cell positions and subcellular markers (especially cytoskeletal [*Priess and Hirsh, 1986*; *Gally et al., 2009*]) in the embryo would help further refine existing models (*Ciarletta et al., 2009*) of embryo elongation.

Currently, building a composite model of neuronal positions and morphological development in the embryo depends on pooling distinct datasets from many independent embryos. Given our experience tracking seam cell nuclei, we next turned our attention to modeling the 4D motion of neurons and neurite outgrowth in the elongating worm embryo as a proof of concept for a neurodevelopmental atlas (*Figure 6A*, *Videos 5,6*). Four of the five embryos used in constructing our seam cell model also had neuronal cell bodies marked with a *pceh-10::GFP* construct; neurons included

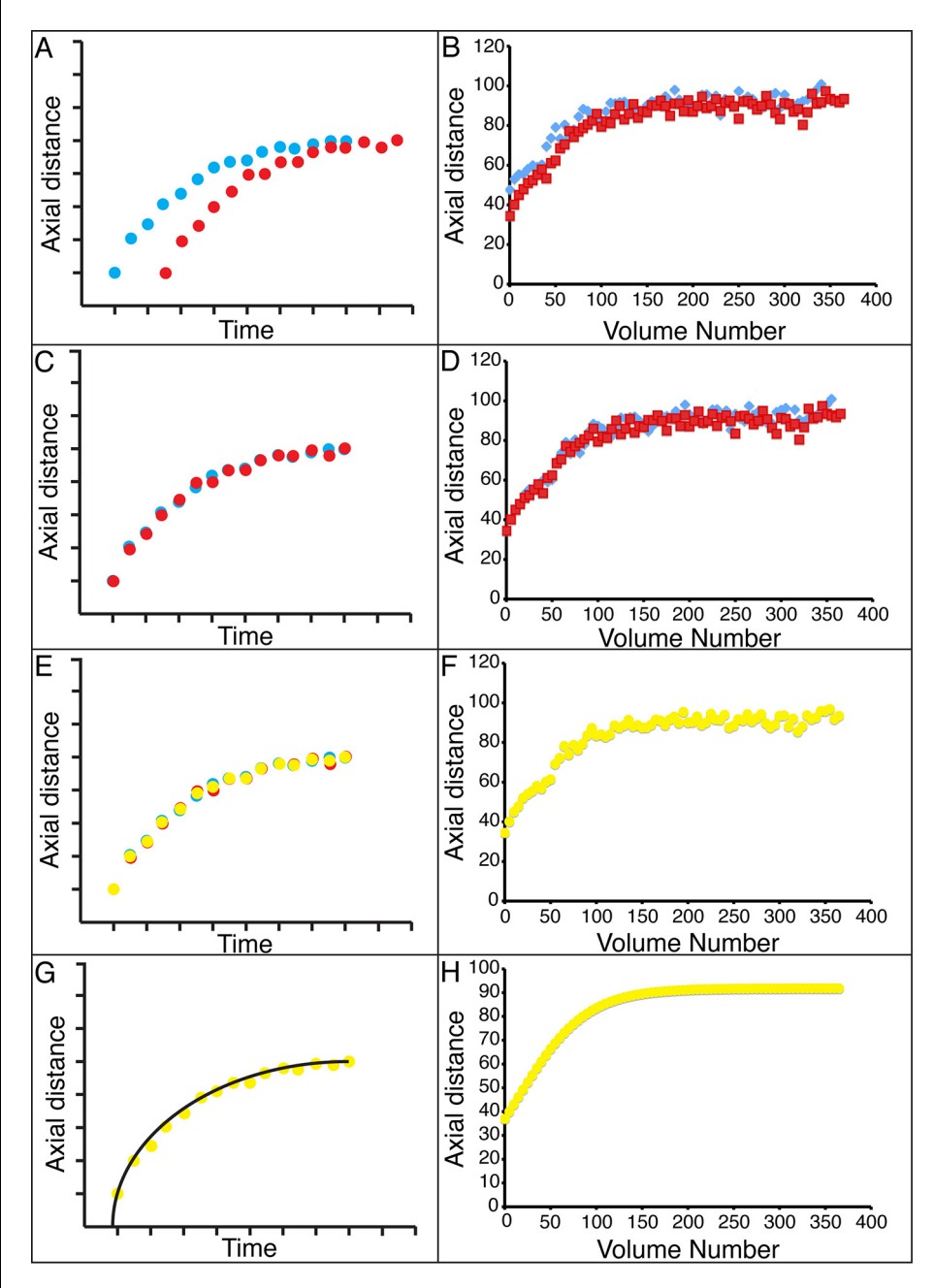

**Figure 4.** Alignment of data from different embryos. (**A,B**) Axial seam cell nuclear trajectories from different embryos are similar in shape, but shifted in time. (**C,D**) Shifting in time aligns the trajectories. (**E, F**) Averaging the shifted trajectories. (**G, H**) Fitting the shifted trajectories. Left graphs: cartoon schematic, Right graphs: data. For clarity, we have shown the shifting, averaging, and fitting process for two embryos, but note that to construct our 'composite' model of seam cell nucleus behavior we have applied the same process to five embryos (see 'Materials and methods' for further details).

The following figure supplements are available for figure 4:

**Figure supplement 1.** Temporal alignment of embryo data.

**Figure supplement 2.** Different fits for axial displacement.

**Figure supplement 3.** Variability in axial distance amongst different embryos.

*Figure 4 continued on next page*

*Figure 4 continued*

**Figure supplement 4.** Fits used in this paper.

AIYL/R, CANL/R, and ALA. We manually annotated the position of these neuronal cell bodies, then temporally aligned, averaged, and fitted the positions as we did the seam cell nuclei (*Figure 4*). The axial motion of these neurons was qualitatively similar to the seam cell nuclei and could be well described by the three parameter logistic function. However, their XY motion appeared different than the seam cell nuclei. For example, the lateral motion of CANs could not be easily described by a simple function, so we used a 50 point smoothing of the averaged data as our 'fit'. The ALA X motion was better described by a 4th degree polynomial than a power law, so we used the former function to fit the data (*Figure 4—figure supplement 4*, *Table 1*). As evident by their axial displacements, ALA and AIYL/R moved similarly to nearby seam cell nuclei (*Figure 6B*). In contrast, CANs moved faster than adjacent seam cell nuclei, suggesting a more 'active' mode of migration (*Figure 6C*). Finally, the motion of ALA and CANs (especially CANL) were considerably more variable between datasets than the seam cell nuclei (*Figure 4—figure supplement 3*, *Supplementary files 2,3,4*). While it is currently unclear whether this variability is strain-dependent or reflects underlying biology, this observation underscores the need to study multiple embryos and assess the degree to which cellular motion is stereotyped in elongating embryos.

To examine neurite outgrowth clearly, we created a two-color strain with GFP-labeled untwisting markers and a *pceh-10::mCh* construct to label neuronal cell bodies and neurites. We observed substantial mosaicism in terms of which cells were labeled from one embryo to the next with this strain. Although neurons were labeled with an extrachromosomal array and a certain degree of mosaicism could be anticipated, labeling differences from one animal to the next hindered our ability to track both ALA and CAN outgrowths. Nevertheless, we were able to obtain two datasets where the ALA neuron was labeled throughout most of our imaging period.

ALA is a single neuron with a cell body located in the dorsal portion of the head; a pair of long neurites extend ventrally from this cell body into the nerve ring, and then turn and extend posteriorly along the lateral nerve cord (*White et al., 1986*). Left and right ALA outgrowths could be readily

**Table 1.** Fitting functions tested for describing axial displacement. Equations are used in *Figure 4—figure supplement 2*. L: length; t: time. Other parameters and their meaning are listed in the table. For all axial coordinates in this paper, a three-parameter logistic function was used.

| Fitting type | Equation | Parameters |
|---|---|---|
| von Bertalanffy | $L = A(1-exp[-B(t-C)])$ | A: upper asymptotic length<br>B: growth rate<br>C: time at which L = 0 |
| Exponential | $L = A-(A-B)exp(-Ct)$ | A: upper asymptotic length<br>B: lower asymptotic length<br>C: growth rate |
| Three-parameter Gompertz | $L = A[exp(-exp(-B(t-C)))]$ | A: upper asymptotic length<br>B: growth rate<br>C: time at which L = 0 |
| Three-parameter logistic | $L = A/[1+exp(-B(t-C))]$ | A: upper asymptotic length<br>B: growth rate<br>C: inflection point |
| Four-parameter Morgan Mercer Flodin | $L = A - (A-B)/(1+(Ct)^D)$ | A: upper asymptotic length<br>B: length at t = 0<br>C: growth rate<br>D: inflection parameter |
| Four-parameter logistic | $L = B + (A-B)/\{1+exp[(C-t)/D]\}$ | A: upper asymptotic length<br>B: lower asymptotic length<br>C: growth rate<br>D: steepness parameter |

**Table 2.** Fitting functions for each cell type. X, Y, Z trajectories were fitted as indicated functions of time (t).' 50-point smoothing' refers to smoothing the input data with a 50-point span, using weighted linear least squares and linear fitting.

| Cell type | X fit | Y fit | Z fit |
|---|---|---|---|
| Seam cell nucleus | Power $X = at^b+c$ | Linear $Y = p1*t + p2$ | Three-parameter logistic $Z = A/(1+exp(-B(t-C)))$ |
| CANR/L | 50-point smoothing | 50-point smoothing | Three-parameter logistic $Z = A/(1+exp(-B(t-C)))$ |
| AIYR/L | $4^{th}$ degree polynomial $X = p4*t^4+p3*t^3+p2*t^2+p1*t+p0$ | Linear $Y = p1*t + p2$ | Three-parameter logistic $Z = A/(1+exp(-B(t-C)))$ |
| ALA ALA xR1/xL1 ALA xR2/xL2 | Linear $X = p1*t + p2$ | Linear $Y = p1*t + p2$ | Three-parameter logistic $Z = A/(1+exp(-B(t-C)))$ |

identified and annotated in both twisted and untwisted embryos (*Figure 6D*, *Figure 6—figure supplement 1*). In modeling the left and right neurite shapes, we simplified them by annotating them as three distinct points (ALA: cell body; AxR1 or AxL1: point at which the neurite turns to extend posteriorly; AxR2/L2: neurite terminus; *Figure 6E*). We then measured the 3D displacements (relative to the nose, as before) of each independent point, shifting, averaging and fitting the data derived from two embryos (as outlined in *Figure 4* and illustrated in *Figure 6—figure supplement 1*), to yield a noise-free representation of the neurite (*Table 1*). Aligning the fitted neurites to our reference embryo allowed inspection of ALA neurite growth in the context of the elongating embryo (*Figure 6A*, *Videos 5,6*), revealing that neurite outgrowth continued to occur for ~240 min after the other cells assumed their final positions at the end of elongation. We also segmented growing ALA neurites at several points in development to demonstrate that straightened images can be used to generate volumetric reconstructions of cell morphology throughout development (*Figure 6—figure supplement 2*.) We are unaware of any other work that has modeled the growth and positions of neurites in the post-twitching embryonic regime (for *C. elegans* or any other model organism).

## Discussion

The *C. elegans* cell lineage is invariant (*Sulston et al., 1983*), and tracking cells in the L1 larva has revealed that cellular positions in post-hatching animals are relatively stereotyped (*Long et al., 2009*). Our work suggests that this positional stereotypy extends to the cells in the late embryo as well. However, we also found that in the case of cells or structures which actively migrate, such as the CAN neurons and ALA neurites, there seems to be greater variability in terms of end position and growth rate. To some extent this is not surprising; as these cells and neurites move longer distances than most other cells, and depend on actively finding their way in a complex environment (as opposed to passive movement in response to elongation), there may be more room for variability in how they travel and reach their destinations.

On a more general level, we also observed variability in the temporal shifts necessary to align each elongating embryo to the reference dataset (embryo 1). Some of this variability may be due to relatively mundane explanations: embryos were at slightly different ages when imaging began, and temperature was moderately controlled (to within 2 to 3°C both during imaging and strain growth). Intrinsic developmental variability, caused by maternal effects or exposure to imaging could also have played a role in the slightly different patterns of development we observed across embryos.

Expanding the work we describe here to other migrating and non-migrating neurons should make clear whether there actually is a difference in positional variability between migrating and non-migrating cells. Adding additional data to our 4D model is conceptually straightforward: strains with distinguishable neurons can be crossed into the untwisting background, untwisted, trajectories of cells and outgrowths fitted, and subsequently registered with previously derived data. 'Filling in' the positions of all neurons and outgrowths in the developing embryo would form the basis of the 4D atlas of neurodevelopment, and could be combined with functional activity mapping and gene

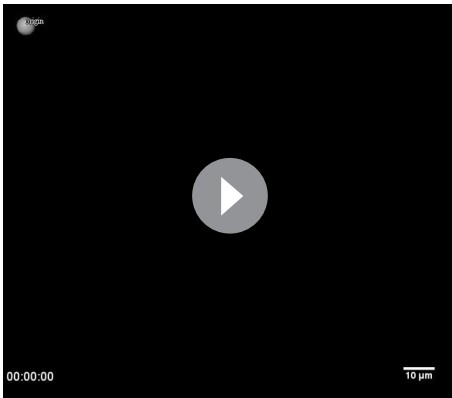

**Video 4.** The same data as in *Video 2*, rendered from a side view.

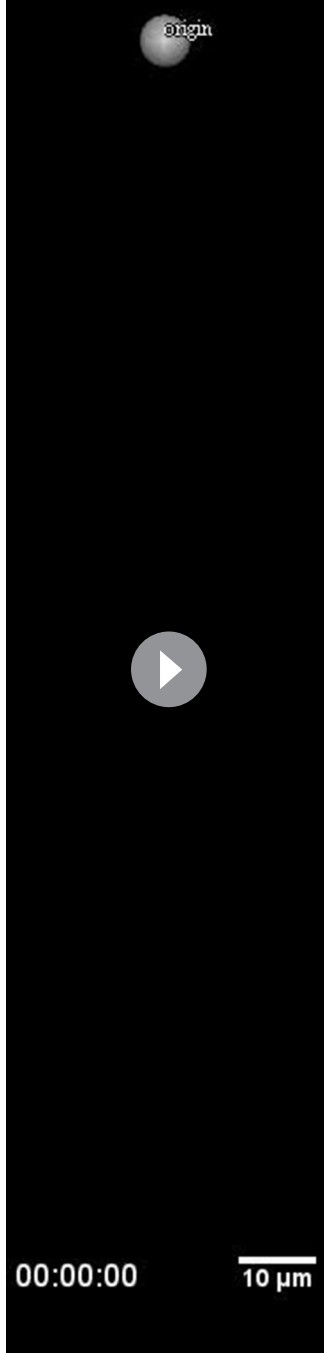

**Video 3.** Rendering of seam cell nuclear positions (gray spheres) in the developing embryo viewed dorsally, from the late 1.5-fold stage until hatching. The positions shown in the rendering are averaged, fitted values derived from five embryos, using the averaging and fitting procedure described in the text; the rendering thus represents a composite, 'best-guess' view as to seam cell evolution in a developing embryo. Times are indicated relative to the first fitted volume, and are 2.5 min apart.

expression data to provide a more comprehensive picture of animal development in late embryogenesis.

Our untwisting and annotation plugin is designed to be flexible, so that it can be applied to most problems involving tracking position and morphology of distinguishable structures in the nematode embryo. The core of the plugin relies on defining the sides of the worm embryo; although our work uses a specific set of markers, we note that any other markers which define the edges of the worm body should also work. The annotation capability is also flexible; as it is based on manual annotation, almost any distinct structure can be annotated. Finally, while the isotropic resolution of the diSPIM is very helpful in resolving fine embryonic detail (*Figure 1—figure supplement 2*), our untwisting algorithm is compatible with other high-resolution imaging methods. For example, we used a super-resolution two-photon instant structured illumination microscope (2P ISIM) (*Winter, 2014*) to image and untwist a bent L2 larval worm, obtaining clear images of this relatively large specimen (*Figure 1—figure supplement 4*, *Video 7*). Our plugin is designed specifically for untwisting nematode embryos, and as such is unlikely to be immediately applicable to other biological systems without substantial modification (we know of few non-nematode systems that have the same vermiform shape and degree of twisting and movement). However, some of the more general concepts we implement, such as the benefit of aligning and pooling information derived different datasets to generate an overall 4D view of development, are likely applicable to more systems than just the worm.

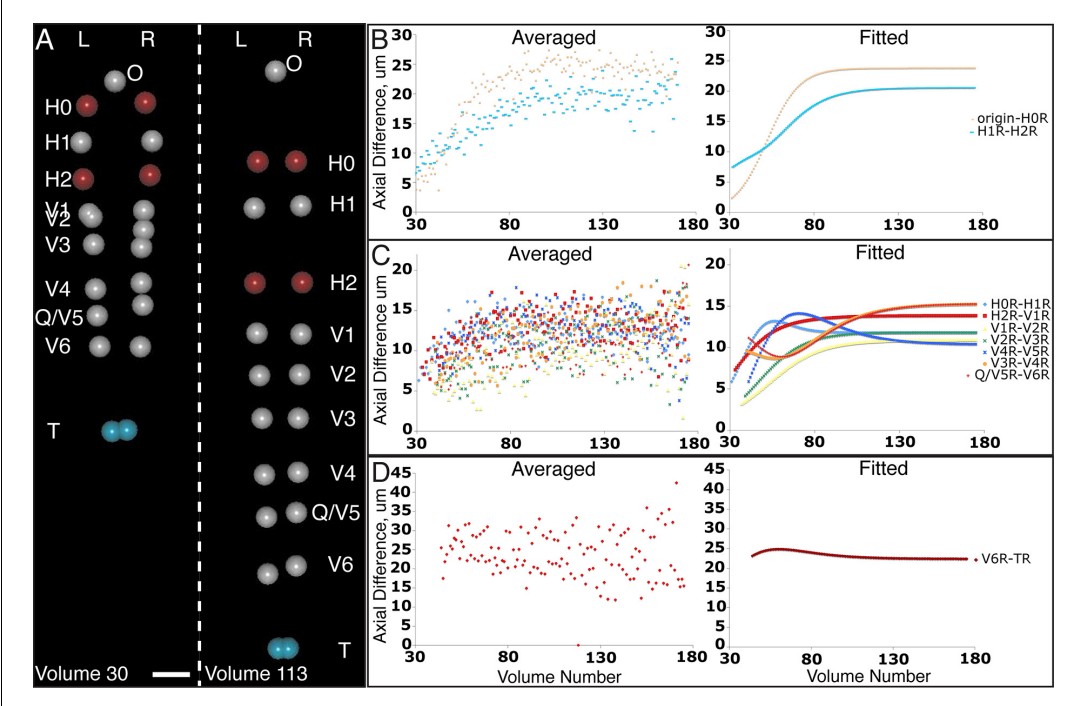

**Figure 5.** Variability in seam cell nucleus axial movement in the elongating embryo. (A) Snapshots of the elongating embryo near start (Volume 30, left) and end (Volume 113) of elongation. Seam cell nuclei volumes are indicated as filled spheres, L/R axes are as indicated, seam cell nuclear identities indicated at the side of each snapshot, as is the origin (nose, 'O'). See also *Videos 3,4*. Scalebar: 10 μm. (B–D) Axial differences over the course of elongation between adjacent seam cell nucleus pairs, sorted into greatest (B), intermediate (C), and least (D) bins, corresponding to red, gray, and blue coloring indicated in (A). Left graphs: raw, averaged data (as in *Figure 4E*, *4F*). Right graphs: fitted data (as in *Figure 4G*, *4H*).

The following figure supplement is available for figure 5:

**Figure supplement 1.** Seam cell nucleus XY movement in the elongating embryo.

Despite the power of our semi-automated approach, several areas for improvement remain. Automated lattice-building assumes the embryo has 20-22 seam cell nuclei on which the lattice is based; in early periods of elongation (especially the 1.5- to 2-fold transition) expression is absent in some seam cell nuclei, requiring manual lattice-building. In addition, time spent in editing automatic segmentation and lattice generation results in ~8 hr of manual work when untwisting an embryo spanning 100–150 timepoints. Fully automated untwisting is not currently feasible, but the development of alternative markers may enable this goal. Second, although the positions of cells and neurites in the growing embryo can be determined with micron-scale precision, and placed in context with their neighbors, additional methods are needed to place the full morphological volume of a given cell within the untwisted embryo. While our data are of sufficient quality to segment such morphology in an untwisted animal (*Figure 6—figure supplement 2*, *Video 8*), the general question about how to combine morphological segmentations from distinct, untwisted embryos remains. New methods developed for pre-twitching embryos may prove useful in this regard (*Santella et al., 2015*).

A more significant and long-term set of technical problems for completing the neurodevelopmental atlas relates to the generation of fluorescent markers and strains that provide sparse, optically resolvable neurons. Most fluorescent strains label multiple neurons that are too close in space and time to be easily resolved – possible strategies to 'separate' these neurons might include 'Brainbow' (*Livet et al., 2007*) (spectral separation of densely labeled neurons) or heat-shock-based approaches (*Halfon et al., 1997*; *Bacaj and Shaham, 2007*) (temporal separation of densely labeled cells). Even if such strains are built, the identity of the resulting neurons will need to be verified. As lineaging (*Bao, 2006*) in *C. elegans* has been carried out to just before twitching

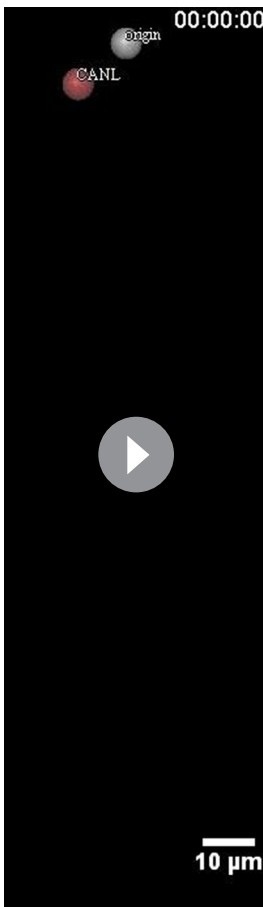

**Video 5.** Rendering of neurons and neurites, in the context of seam cell nuclei shown in *Videos 2,3*. As in these videos, all positions are averaged, fitted values derived from multiple embryos. View is from dorsal perspective. Red spheres represent CAN cell bodies, yellow spheres represent AIY cell bodies, and blue spheres and lines correspond to ALA and its neurites. ALA and AIY cell bodies appear to closely track neighboring seam cells during elongation, while the CAN neurons actively migrate. ALA neurite outgrowth starts toward the end of elongation and continues after most other morphological changes have ceased. Times are indicated relative to the first fitted volume, and are 2.5 min apart.

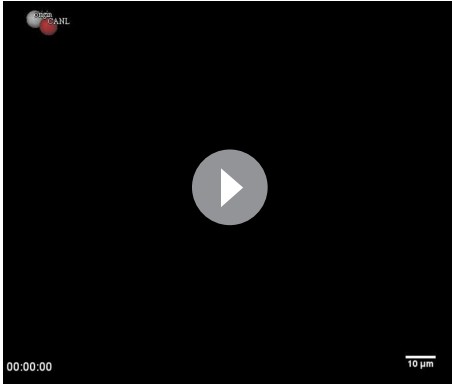

**Video 6.** The same data as in *Video 4*, rendered from the side.

begins (*Giurumescu et al., 2012*), in principle neurons can be identified by matching early expression to lineage data. If expression turns on after twitching, lineaging would also need to be extended into the post-twitching regime. Such 'deep lineaging', or tracking the coordinates of all nuclei through twitching would be a valuable and complementary effort to untwisting. Finally, we note that the expression pattern of fluorescent proteins within individual neurons could be further optimized. For almost all strains (except DCR4209 which contained membrane-targeted mCherry), fluorescent proteins were expressed cytoplasmically. An improved strategy would combine such cytoplasmic labeling with membrane targeting, better filling out very thin neuronal outgrowths that otherwise might be missed due to low expression; a similar strategy was adopted in super-resolution microscopy to trace thin neurites (*Lakadamyali et al., 2012*).

## Materials and methods

### Strains

Nematode strains were kept at 20°C, and grown on NGM media plates seeded with *E. coli* OP50. The untwisting strain is SLS1 *xnIS17 [dlg-1::GFP + rol-6]; wIS51 [SCM::GFP]*. Strains used to construct SLS1 were FT63 [*xnIS17 dlg-1::GFP + rol-6*] (*Totong et al., 2007*) and JR667 [*wIS51 SCM::GFP*] (*Terns et al., 1997*; *Koh and Rothman, 2001*). Strains were crossed together to generate an animal containing these transgenes. Strains imaged for the paper include SLS1, DCR4209, and DCR4221. Strain DCR4209 contained the following transgenes: *olaex2457 [P.ceh-10::mCh-PHd* (25 ng/µL) + *unc122::RFP* (30ng/µL)]; *xnIS17 [dlg-1::GFP + rol-6]; wIS51 [SCM::GFP]*. To create *olaEX2457*, 4132 bp upstream of the transcriptional start site were isolated using the following promoters: Forward AGC TCC TGC ACT CTT CTG ATC; Reverse CAC AAG AGA AAA GTG GCT GCT TAT C. Strain DCR4221 contained the following transgenes: *lqIS4 (Wenick and Hobert, 2004) [ceh-10promA::GFP]; xnIs17 [dlg-1::GFP + rol-6]; wIs51 [SCM::GFP]*. Detailed subcloning information for *olaex 2457* can be provided upon request.

## Sample preparation

As previously described, worm samples were prepared for diSPIM (*Wu et al., 2011*; *Bao and Murray, 2010*; *Kumar et al., 2014*): adult animals were placed in buffer and cut to liberate embryos, embryos transferred to poly-L-lysine-coated coverslips in the diSPIM imaging chamber, and imaged once they reached the bean-to-comma stage of embryonic development.

## Data acquisition

All data were acquired on either a first-generation diSPIM (*Wu et al., 2013*) or a more recent fiber-coupled version (*Kumar et al., 2014*). Dual-color data were taken sequentially (first the 488-nm excitation for the GFP channel, and then 561-nm excitation for the mCherry channel) in a plane-by-plane (5 ms GFP collection, 5 ms mCherry collection per axial position in the embryo) fashion. Given 50 planes per view, and two perpendicular views, this resulted in an acquisition time of 1 s per 2-color diSPIM volume. For most datasets in this paper (embryos 2-8, as referred to elsewhere in the text),

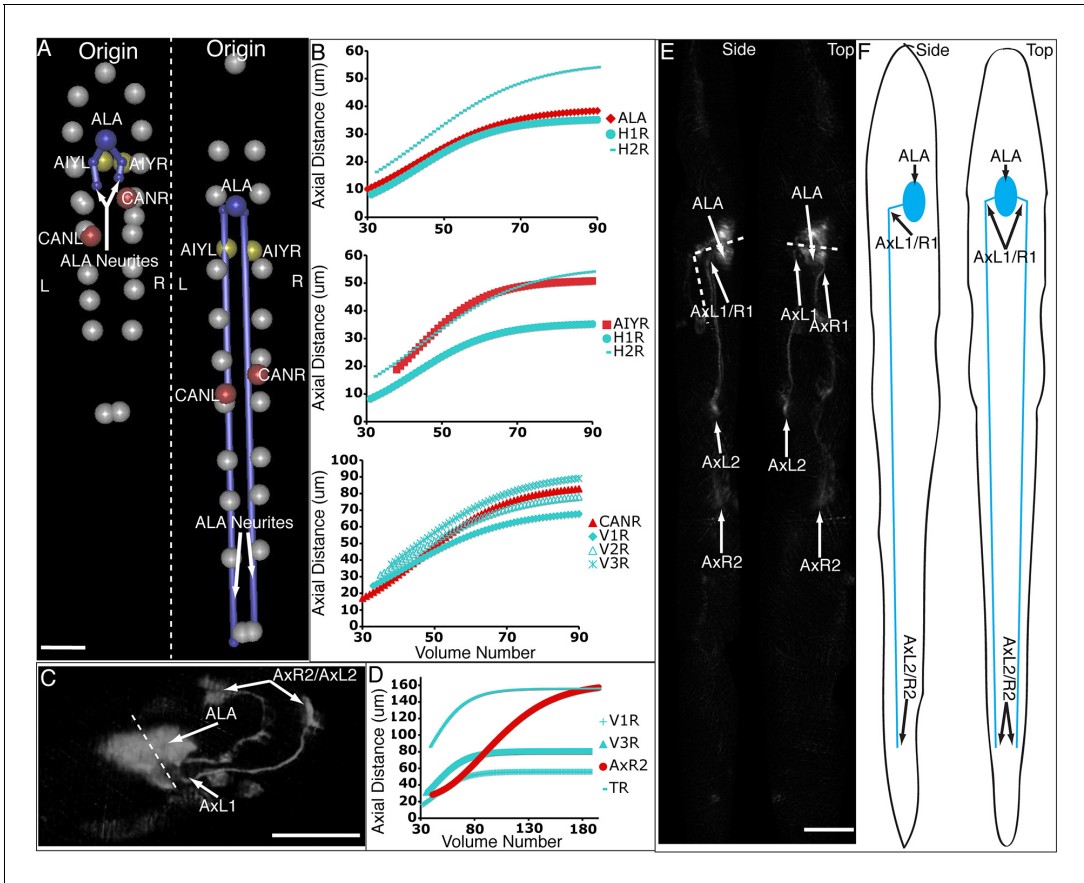

**Figure 6.** Neurons and neurites in the developing embryo. (**A**) Early (left) and late (snapshots) in the elongating embryo. Gray spheres: seam cell nuclei; ALA cell body: blue sphere; ALA neurites: blue lines; AIY cell bodies: yellow spheres; CAN cell bodies: red spheres. Compare to *Videos 5,6*. (**B**) ALA (top), AIYR (middle), and CANR (bottom) axial trajectories (red curves) in relation to neighboring seam cells (blue curves). ALA and AIY cells maintain their relative position with respect to the rest of the elongating body, while CANs migrate faster than neighboring seam cells. (**C**) ALA cell body and neurite in the twisted embryo, highlighting morphological features (ALA: ALA cell body; AxL1/R1: junction between ventral and posterior neurite extension; AxL2/R2: posterior tip of the ALA neurites). (**D**) Axial trajectory of ALA neurite tip in relation to indicated seam cells. (**E**) Top and side models of ALA in untwisted reference frame, indicating neurite bend and terminus. Compare to *Figure 6—figure supplement 2*.

The following figure supplements are available for figure 6:

**Figure supplement 1.** Shifting, averaging and fitting procedures for modeling the ALA neurite.

**Figure supplement 2.** Segmentation of neurons and neurites in the untwisted embryo.

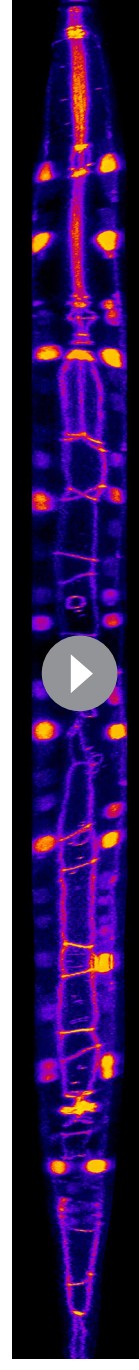

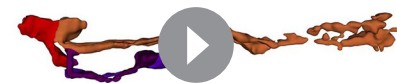

**Video 8.** Rotating three-dimensional view of the segmentation shown in *Figure 6—figure supplement 2*. The volume was segmented and rendered in Imaris.

single-color volumes were acquired every 5 min, but for one datastet (embryo 1), single-color volumes were acquired every 2.5 min. Dual-color acquisitions were used to track ALA neurite outgrowth (embryos 7 and 8). Acquisition code, written in LabVIEW, is available at http://www.wormguides.org/dispim/dispim-downloads.

For 2P ISIM imaging, we used 900-nm excitation and two 680-nm short-pass filters (Semrock, FF01-680/SP-25) in our emission path to filter illumination light. L2 larvae of strain SLS1 were immobilized with 50 mM levamisole (Sigma-Aldrich; St Louis, MO) and imaged on an agarose pad sandwiched between two #1.5 coverslips. Volumetric images of the entire specimen were acquired by manual XY translation of the stage between fields of view. Each raw frame was acquired in 200 ms; data used in this paper were derived by averaging six raw frames per axial position. Axial positions were spaced 0.333 µm apart. Individual 3D image stacks were stitched and overlaid to reconstruct the entire L2 stage worm using a custom plugin developed for MIPAV (available online at www.cit.nih.gov/mipav). After stitching, the reconstructed L2 stage worm volume was further processed with 40 iterations of Richardson-Lucy deconvolution.

## Shifting and averaging trajectories derived from different embryos

Cells from different embryo datasets exhibited qualitatively and quantitatively similar trajectories, so we aligned and then combined them to generate averaged, noise-free trajectories. First, coordinate trajectories (X, Y, or Z positions (*Figure 3I*) as a function of time) were 'cleaned'

**Video 7.** Rotating view of an untwisted L2 worm. The image was imported into ImageJ and the Magenta LUT was applied to the stack. The volume shown here corresponds to the untwisted volume in *Figure 1—figure supplement 4*.

to remove obvious outliers, or to linearly interpolate gaps in the raw data (*Figure 3—figure supplement 3*). Second, the axial ('Z') coordinate of each cell was fitted to a three parameter logistic function (*Table 2*) using Growth II (Pisces Conservation) or MATLAB (Mathworks) software, as this function provided a better fit than other three parameter growth curves, and did not require careful

tuning of initial parameter values, as did the four parameter growth curves we tested (*Figure 4—figure supplement 4*, *Table 1*). Third, we aligned datasets from embryos 2-5 (volumes recorded every 5 min) to embryo 1 (volumes recorded every 2.5 min), by (i) determining the inflection time point ('C' in *Table 1*, *2*) for each cell's fitted axial position and (ii) shifting the data an integral number of time points so that the inflection time points from embryos 2-5 agreed with the inflection point for embryo 1. For example, for the data shown in *Figure 4—figure supplement 1* for seam cell V3R nuclei, embryo 1 had inflection point 42.6, and embryo 4 had inflection point 37.7, so the V3R trajectory for embryo 4 was shifted 42.6-37.7 = 5 timepoints to the right, to match the trajectory for embryo 1. The same integer time point shift was then applied to the corresponding 'X' and 'Y' coordinate trajectories for each cell. Fourth, after shifts were applied, coordinate trajectories were averaged. Finally, to generate noise-free trajectories, the average trajectories were fitted (functions chosen for the fits are shown in *Figure 4—figure supplement 4* and *Table 1*).

To examine the degree to which embryo positions agreed after the shifting procedure, we computed SD between embryo positions at each time point (*Supplementary file 2*). With the exceptions of the CAN neurons, the X and Y positions of cells were stereotyped to within 2 μm, and the Z positions within 10 μm.

## Validating fits for each embryo

The majority of cells' coordinate trajectories were well described by power (X coordinate), linear (Y coordinate), and three-parameter logistic (Z coordinate) functions (*Figure 4—figure supplement 4*, *Tables 1*, *2*, *Supplementary file 3*). However, two cell types, CAN and AIY, were not well described by any of the common fitting functions we surveyed (e.g. power, exponential, Gaussian, rational functions). For these cells, we instead used 50 point smoothing (for CAN X and Y coordinates) or a quartic polynomial function (for AIY X coordinates) to reduce noise in the shifted, averaged trajectories. To estimate how well the curve fitting described the averaged trajectories, we calculated the absolute differences between averaged and fitted coordinates at each time point, and then calculated the means and SD of these differences across time. These data are recorded in *Supplementary file 3* as $\mu_{avg\text{-}fit,\ time}$ and $\sigma_{avg\text{-}fit,\ time}$. We also computed the average over all seam cell nuclei of these average differences to generate $<\mu_{Xavg\text{-}Xfit,\ time}>_{seam\ cell}$; $<\mu_{Yavg\text{-}Yfit,\ time}>_{seam\ cell}$; and $<\mu_{Zavg\text{-}Zfit,\ time}>_{seam\ cell}$ resulting in values of 0.5 μm, 0.6 μm, and 3.7 μm. In XY, similar average statistics were found for all cell types. In the Z coordinate, CANL stood out as more variable, as its $\mu_{Zavg\text{-}Zfit,\ time}$ was 10.2 μm (with a corresponding SD of 9.3 μm). We suspect the deviation between CANL data and fit arises more from the inherent variability with CAN cells (*Supplementary file 2*) than inherent problems with the fitting function choice.

## Population statistics

In several locations, we report population averages taken across some combination of seam cell nuclei, time, or embryos. We use μ and s to denote mean and SD, and $<X>_Y$ indicates an average of quantity X, across Y. For example, $<\mu_X>_{embryo}$ stands for the population average across embryos, of mean X coordinate positions (each derived from an individual embryo).

For untwisting control measurements, we measured the difference between twisted and untwisted volumes for various distance metrics (between seam cells and along the pharynx). For each embryo, we computed the mean difference $\mu_{Difference,time}$ and standard deviation $\sigma_{Difference,\ time}$ across time, and averaged these quantities to calculate a population $<\mu_{Difference,\ time}>_{embryo}$ and population $<\sigma_{Difference,\ time}>_{embryo}$ across embryos.

To estimate inter-embryo and inter-seam cell nuclei positional (X, Y, and Z coordinates) variability over elongation, we shifted data from embryos until they overlaid in time, and next computed the SD between embryo positions at each aligned timepoint. Mean standard deviations $<\sigma_X>_{time}$; $<\sigma_Y>_{time}$; and $<\sigma_Z>_{time}$ over all timepoints were calculated, and are reported in *Supplementary file 2*. To compute $<<\sigma_X>_{time}>_{seam\ cell}$; $<<\sigma_Y>_{time}>_{seam\ cell}$ and $<<\sigma_Z>_{time}>_{seam\ cell}$, we averaged mean SD across the 20 seam cell nuclei.

## Supplementary datasets

In accordance with *eLife* policy, we have made our raw annotation data and quality control measurements available: *Supplementary file 4* contains the 3D positions of seam cell nuclei, neurons, and

growing ALA axons. These data were used in *Figures 3–6*. Data are provided before outlier removal, shifting, and fitting. *Supplementary file 5* contains the quality control measurements (distances between seam cell nuclei in the H0 and T pairs before and after untwisting, and pharyngeal contour lengths before and after untwisting) used to generate *Figure 2*.

## Acknowledgements

We thank Joseph Joy for early inspiration and encouragement, Nish Pandye (CIT, NIH) for initial work on the untwisting plugin, and Andrea Zou and John Joseph for assistance with untwisting. We thank Jing Chen, Zhanghan Wu, and Jian Liu for helpful discussion on the biomechanics of embryo elongation. We also thank the Research Center for Minority Institutions program and the Institute of Neurobiology at the University of Puerto Rico for providing a meeting and brainstorming platform. This work was partially conducted at the Marine Biological Laboratories at Woods Hole, under a Whitman research award (to DAC-R, ZB and HS). This work was supported by the Intramural Research Programs of the NIH National Institute of Biomedical Imaging and Bioengineering, the Center for Information Technology, NIH grants U01 HD075602 and R24OD016474, National Natural Science Foundation of China grants 61427807, 61271083, 61525106, and by Natural Science Foundation of Zhejiang grants LR12F03001. We acknowledge Dr. Jeffrey Simske for sharing strains. Some strains were also provided by the CGC, which is funded by NIH Office of Research Infrastructure Programs (P40 OD010440). The NIH, its employees, and officers do not endorse or recommend any commercial products, processes, or services.

## Additional information

### Funding

| Funder | Grant reference number | Author |
|---|---|---|
| National Institute of Biomedical Imaging and Bioengineering | | Ryan Patrick Christensen<br>Yicong Wu<br>Peter W Winter<br>Hari Shroff |
| Center for Information Technology | | Alexandra Bokinsky<br>Evan McCreedy<br>Matthew McAuliffe |
| National Natural Science Foundation of China | 61427807 | Huafeng Liu |
| National Natural Science Foundation of China | 61271083 | Huafeng Liu |
| National Natural Science Foundation of China | 61525106 | Huafeng Liu |
| Natural Science Foundation of Zhejiang Province | LR12F03001 | Huafeng Liu |
| Marine Biological Laboratory | Whitman Research Award | Daniel A Colón-Ramos<br>Zhirong Bao<br>Hari Shroff |
| National Institutes of Health | UO1 HD075602 | Daniel A Colón-Ramos<br>Zhirong Bao<br>Hari Shroff |
| National Institutes of Health | R24OD016474 | Daniel A Colón-Ramos<br>Zhirong Bao<br>Hari Shroff |

The funders had no role in study design, data collection and interpretation, or the decision to submit the work for publication.

### Author contributions

RPC, YW, Conception and design, Acquisition of data, Analysis and interpretation of data, Drafting or revising the article; AB, Conception and design, Analysis and interpretation of data, Drafting or

revising the article ; AS, MG, DAC-R, ZB, HS, Conception and design, Analysis and interpretation of data, Drafting or revising the article; JM-S, AK, PWW, Acquisition of data, Analysis and interpretation of data, Drafting or revising the article; IK, EM, MM, WM, Conception and design, Drafting or revising the article; NT, Analysis and interpretation of data, Drafting or revising the article; HL, Conception and design, Analysis and interpretation of data

## Additional files

### Supplementary files

• Supplementary file 1. Tutorial for use of the WormUntwisting automated lattice-building plugin.

• Supplementary file 2. Deviations between embryo datasets. For each cell studied in this paper, data from 5 embryos were shifted as discussed in the text, and the standard deviations between embryo positions at each timepoint computed. Mean standard deviations ($<\sigma>$) and the maximum standard deviation (Max($\sigma$)) over all timepoints are recorded above. For x and y coordinates, the majority of embryo positions are within 2 µm. For z coordinates, most embryo positions lie within 10 µm of each other, with the exception of CANL. See also *Figure 4—figure supplement 3* for representative embryo trajectories. For most data displayed here, at least three embryo datasets were used in generating these values. For three datasets (red italics), only two embryo datasets were compared.

• Supplementary file 3. Deviations between fits and averaged data. For each cell studied in this paper, the absolute differences between averaged coordinates and fits were computed at each time point. The means and standard deviations of these differences over time, in µm, are recorded in the table above. For x and y coordinates, the majority of fitted points lie within 1.5 µm of the averaged data, regardless of cell type. For z coordinates, the majority of fitted points lie within 7.5 µm of the averaged data, with the exception of CANL.

• Supplementary file 4. Raw annotation data for seam cell nuclei, neuronal cell bodies, and ALA neurites. Supplementary data file 4 contains raw annotation data generated by the untwisting algorithm for the 20 seam cell nuclei; CAN, AIY, and ALA cell bodies; and ALA neurites. Each sheet contains positional information for one cell, broken up by embryo dataset. Embryo datasets are labeled in the form Embryo_#_X_minutes, where # corresponds to the number assigned to the dataset (1–8) and X represents the imaging frequency (between volumes) in minutes. For each embryo dataset, the volume numbers and X, Y, and Z-positions of the cell or neurite in that volume are listed.
The data are provided in raw form, after sorting by embryo, cell, and volume but before cleaning, shifting, and fitting. For some volumes annotation information was not captured, usually due to errors in the untwisting process; for these volumes the spreadsheet entries are left blank. Additionally, there is unconstrained rotation around the midline in most datasets, which can cause X and Y-values to switch between positive and negative sign. The canonical orientation of the embryo for this paper is for cells on the right side (R) of the animal to have positive X-values and Y-positions located dorsal to the midline to have positive values; in volumes where the XR values are negative the sign should be changed, as well as the corresponding sign for the YR, XL, and YL values for that volume. Z-measurements are insensitive to this rotation. All annotations are in µm.

• Supplementary file 5. Quality control measurements. The data provided in this supplementary data file correspond to the quality control measurements used to generate *Figure 2* and *Figure 2—figure supplement 1*. The data are sorted by embryo, volume, and measurement type. Embryos are named in the form Embryo_#_X_minutes, where # corresponds to the number assigned to the dataset (1–8) and X represents the imaging frequency (between volumes) in the dataset. All data are listed in µm.

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

**Appendix**

## Appendix, further details on lattice building, worm modeling, and untwisting

The untwisting algorithm is a semi-automatic process involving automatic segmentation of the seam cell nuclei and automatic lattice-building combined with user-interactive reviewing and editing phases. This procedure results in a lattice that bends and twists in the 3D volume, capturing the structure of the worm embryo. The lattice sets the parameters for generating the worm model, which in turn defines how the worm is untwisted.

## Pipeline overview

The pipeline processes the series of volumes in batch mode, then presents the results to the user for viewing and editing. The user can quickly step through the results making changes as necessary then proceed with the next batch process. The pipeline automatically switches to the next step in the process once the current step has completed. Steps are:

1. Automatic seam cell nuclei segmentation

2. Review and edit seam cell nuclei (the user adds an optional nose marker or seam cell segmentations)

3. Automatic lattice-building

4. Review and edit lattice (the user adds optional lattice points to better follow the curvature of the worm)

5. User interactively places annotations marking points of interest

6. Automatic model-building and untwisting

7. Review straightened volumes

## Automatic seam cell segmentation

Several standard segmentation and clustering algorithms have been tried; to date the most accurate segmentation algorithm clusters regions of uniform intensity into seam cell nuclei. The algorithm searches for clusters with diameters in the range of 2.5–4.8 microns. This criterion prevents regions of the worm digestive tract from being labeled as a seam cell nucleus - even though the high intensity and near-spherical appearance of the digestive tract matches the seam cell nuclei, the digestive tract is less uniform throughout its structure.

There are several challenges to producing an accurate segmentation of the seam cell nuclei. The algorithm must segment nuclei with very different intensity threshold values, sizes and shapes. Not all seam cell nuclei are spherical due to motion blur. The algorithm must distinguish between seam cell nuclei and other features of the worm similar to the seam cell nuclei, for example the valves in the worm's digestive tract. Often the nose of the worm is labeled a seam cell nucleus while the 10th pair of seam cell nuclei, because they appear much fainter and are closer together than all other pairs, are combined into a single marker or simply not segmented at all. Accurate segmentation is complicated by division of the 8th pair of seam cell nuclei (QV5) at the later stages of the embryo's growth. During subdivision the QV5 pair is difficult to detect, and they may divide at different times, which causes the algorithm to detect 21 cells instead of 20 or 22 (*Appendix 1—figure 1*). These factors imply that the user must review and edit the seam cell nuclei placements.

The user experience during the editing phase in turn influences the segmentation algorithm. Users have found that it is quicker and easier to add missing seam cell nuclei than to delete extra seam cell nuclei. This feedback caused us to modify the segmentation algorithm, erring on the side of more accurate segmentation that detects fewer seam cell nuclei, rather than labeling extra regions as seam cell nuclei in the attempt to detect all 20 (*Appendix 1—figure 1*).

## Review and edit

When the automatic seam cell nuclei segmentation is complete, the user can view each file and edit the segmentations (*Appendix 1—figure 2*). The user also has the option to add an additional point to mark the nose of the worm, which increases the accuracy of automated lattice-building (*Appendix 1—figure 3*). Editing is accomplished by clicking in the volume view pane with the mouse.

Selecting a 3D position in the volume with the 2D mouse input device is accomplished by mapping the mouse position to a 3D position in space using the maximum intensity in the volume that falls under the mouse location. The model transformations (translations, rotations), viewing transformation (zoom), and projection transformations are used to generate a 3D ray originating at the 2D mouse position. The ray passes through the 3D volume along the viewing direction. The 3D volume position (voxel) along the ray with the maximum intensity determines the selected point in 3D.

Because the automatic lattice-building algorithm takes as input either 20 or 22 unordered seam cell nuclei, the user is encouraged to ensure that after editing the correct number of seam cell nuclei are labeled. To help the user achieve this goal, the color of the seam cell nuclei markers change to indicate if the target number has been reached. All seam cell nuclei are colored yellow when there are either 21 seam cell nuclei marked or fewer than 20 nuclei marked (*Appendix 1—figure 1*). If exactly 20 or 22 seam cell nuclei are marked they are colored green (*Appendix 1—figure 2*). Anytime more than 22 seam cell nuclei are labeled all are colored red. This feature provides a visual clue to the user if there are too few or too many seam cell nuclei, and that editing is not complete.

The following example images show results from the automatic seam cell nuclei segmentation algorithm followed by the edited segmentations:

## Automatic lattice-building

The automatic lattice building algorithm generates a lattice representation of the worm. The algorithm takes as input a set of up to 22 unordered points in the 3D volume, as well as an (optional) nose point. The algorithm automatically generates lattices based on the relative positions of those points and sets of threshold values that were determined empirically from hand-generated lattices, i.e. from priors determined directly from measurements on untwisted worm volumes. The output is the 5 highest-rank lattices where the lattices are evaluated based on minimizing the overall curvature of the lattice as well as minimizing the amount each lattice self-intersects. The algorithm for automatically generating lattices from the collection of 3D points is described in detail below.

Given a set of up to 22 unordered seam cell nuclei positions in 3D, the algorithm generates pairs of seam cell nuclei and outputs an ordered sequence of those pairs. The ordered sequence forms a lattice from the head of the worm to the tail of the worm. If the nose of the worm is included in the set of points it is incorporated into the lattice and used at the evaluation and ranking stages.

The algorithm works in a stepwise fashion, first identifying the T seam cells and then working nosewards from them to build potential lattices. Lattices are checked both during construction and after completion for compliance with a series of parameters experimentally determined from hand-built lattices; lattices that pass all checks are displayed to the user for final selection and editing. The steps in the lattice-building process are described in more detail below.

**Step 1:** Find and label all possible 10th pairs. The 10th seam cell nuclei pair is located at the tail of the worm. It is distinct from the 9 other seam cell nuclei pairs as it is much closer together. The minimum and maximum distances between seam cells for the 10th pair are set to 1 and 5 microns. In contrast, the minimum and maximum distances for all other pairs are set to 5 to 15 microns.

Pairs of points that fall within the minimum and maximum thresholds for the 10th pair are evaluated using a mid-point test. The test calculates the mid-point between the two seam cell nuclei and if there is another seam cell nucleus that is closer to the mid-point than the potential pair, the pair is ruled out. If the pair passes all mid-point tests it is labeled as a potential 10th pair.

**Step 2**: For each potential 10th pair, build all possible sequences starting at the 10th pair and work backward to the head of the worm.

a. The first step in building sequences from a given 10th pair is to generate the set of all potential pairs with the remaining points. Pairs must fall within the minimum and maximum threshold values of 5-15 microns. In addition each pair must pass the mid-point test, ensuring that no other point is closer to the mid-point of the pair.

b. For a given set of pairs, the set is searched to determine if any point occurs only in one pair. This pair is deemed an 'essential pair', as not including it would mean excluding the point found only in that pair. For each essential pair the remaining pairs are examined and all pairs with the partner of the unique point are eliminated. This process of elimination can generate other essential pairs so this step is repeated until no more essential pairs are found.

c. Once the pair set is determined, sequences are built from the 10th pair until a complete lattice is formed. If at any point in the lattice-building process the lattice in progress fails to meet criteria it is eliminated from the set of potential lattices.

Given a sequence of pairs in a lattice, a new pair is added to the lattice if the following criteria (determined from analysis of hand-built lattices) are met:

i. The angle between the left-edge and right-edge from the last pair in the sequence to the current pair (the amount of twist) is less than 90 degrees.

ii. The difference in length between the left-edge and right-edge sequences is less than 12 microns.

iii. The distance from the mid-point between last pair and the mid-point of the current pair is greater than 4 microns and below 30 microns.

iv. The distance between the last left-edge point and the current left point is greater than 4 microns and less than 25 microns.

v. The distance between the last right-edge point and the current right point is greater than 4 microns and less than 25 microns.

**Step 3**: When a sequence reaches a length of 10 (or 11) pairs it is evaluated. Sequences that fail to reach the target length are eliminated. A sequence is accepted if it meets the following criteria:

a. For each pair in the lattice no other point is closer to the pair mid-point (***Appendix 1—figure 4***).

b. The average width of the first 4 lattice pairs is wider than the average width of the last 5 lattice pairs.

c. The maximum width occurs in the first 4 lattice pairs.

d. The curvature of the lattice is between 2 (540 degrees of bend) and 4 folds.

e. The total length of the lattice, calculated by summing the mid-point distances is greater than 100 microns and less than 140 microns.

f. The distances between the mid-points of lattice pairs fall within threshold values, with a minimum distance of 5 microns and a maximum distance of 25 microns.

**Step 4:** Lattices are ranked to determine the top 5 lattices. Ranking is determined by minimizing the following two values:

a. Lattice curvature, which is a measure of the total amount of bend as calculated by measuring the angle from one mid-point on the lattice to the next.

b. Amount of lattice self-intersection, which is approximated by intersecting the bounding boxes formed around sequential lattice pairs.

As an example, the algorithm starts with a set of 20 unordered seam cell nuclei positions in 3D. Two pairs were categorized as being a possible 10th pair. For the first of the possible 10th pairs, the remaining 18 seam cell nuclei, which unconstrained would generated a set of (18!) possible pairs, instead produced 100 pairs that fell within the minimum/maximum distance threshold, of which only 31 met the mid-point test. After eliminating pairs based on the 'essential pair' criteria only 27 potential pairs remained. For the second of the possible 10th pairs, with 18 remaining seam cells only 104 possible pairs met the minimum/maximum distance threshold and of those only 33 pairs satisfied the mid-point test. Eliminating pairs based on the 'essential pair' criteria further reduced the pool of potential pairs to 28. 27 choose 9 is 4,686,825 lattices and 28 choose 9 is 6,906,900 lattices; too many to evaluate. However, automatically eliminating lattices during the building process reduced this number to 22,916 lattices that reached the target length of 10 pairs. Of those only 21 lattices passed the evaluation phase. After ranking the 21 lattices, the correct lattice was listed at the top-ranked lattice (*Appendix 1—figure 5*).

## Review and edit lattice

During the review and edit phase the user steps through the selected volumes. For each volume up to 5 of the highest-ranked lattices are presented to the user, enabling the user to select the lattice that best fits the data. In some cases the algorithm finds less than 5 lattices and presents all those found. The user has the option to modify the selected lattice by adjusting the placement of points in the lattice or by adding points to the lattice so that the model better matches the worm. Once the user has finished editing the lattice it is saved for future reference (*Appendix 1—figure 6*).

## Building the lattice manually

When the automatic lattice-building algorithm fails, the user has the option to build the entire lattice by hand, marking the points of the lattice with the mouse.

The user creates a lattice for a worm volume interactively in the volume view of the plugin. The volume view enables the user to rotate the volume in 3D to view it from different angles, or to view the volume at different scales. The user selects points in the 3D volume by clicking with the mouse.

The user starts at the head of the worm and interactively selects points in the 3D volume as left-right pairs, building the lattice as they go. The left-right markers visible along the sides of the worm (consisting of fluorescently-labeled seam cells and junctions between epidermal calls) are used to build the lattice. Because the volume appears mostly transparent except for the seam cell markers, the left-right markers are easy to select in 3D. The user can also interactively move points in the lattice after they are placed to better fit the data. Undo and redo features are also part of the lattice-building interface.

There are 10 pairs of seam cells along the left and right sides of the worm body. While these cells act as the primary markers when building the lattice, they are spaced too far apart to capture every bend in the worm. Labeled contacts between epidermal cells provide another set of markers which can be selected by the user to create additional pairs. This enables the user to better define the curve of the worm. *Appendix 1—figures 7–9* show lattice building in progress.

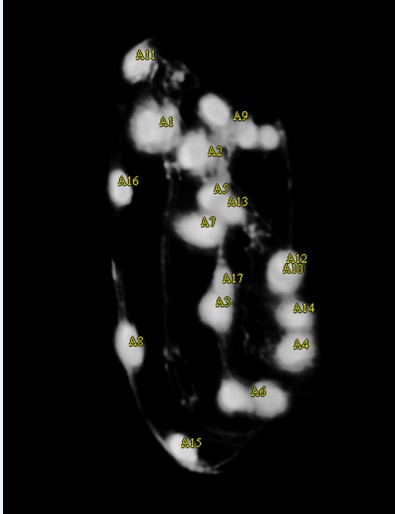

**Appendix 1—figure 1.** Output of the automatic seam cell nucleus detection algorithm shown before editing starts. The markers are yellow, indicating that fewer than 20 seam cell nuclei have been labeled.

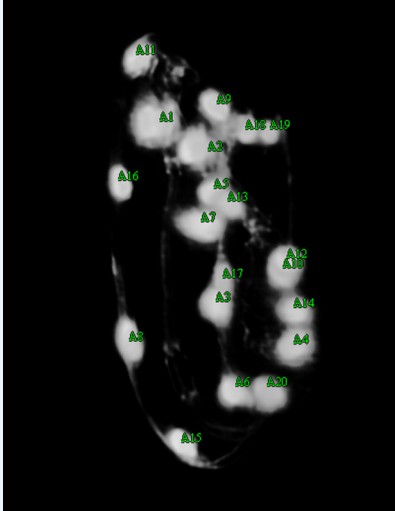

**Appendix 1—figure 2.** User editing. The user shifts seam cell nucleus #9 over, adds markers for

both of the 10<sup>th</sup> seam cell nuclei, shifts nucleus #6 over and adds nucleus #20. There are now 20 seam cell nuclei marked, as indicated by the green color.

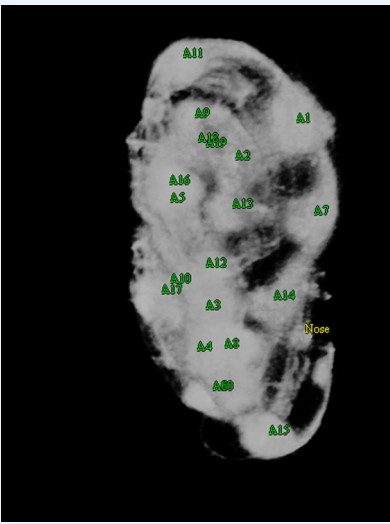

**Appendix 1—figure 3.** Nose labeling. The user has increased the opacity of the volume to better enhance the appearance of the nose, now labeled in yellow.

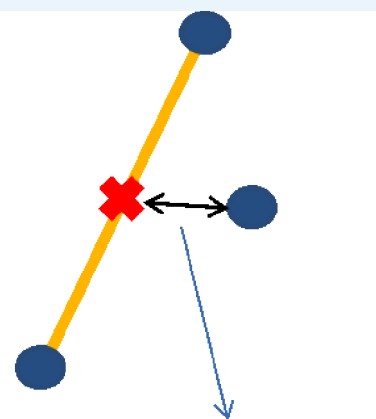

**Appendix 1—figure 4.** Pairing quality control. A potential pair is found, with the mid-point marked in red. A third seam cell nucleus is found closer to the mid-point than the pair, invalidating the pair.

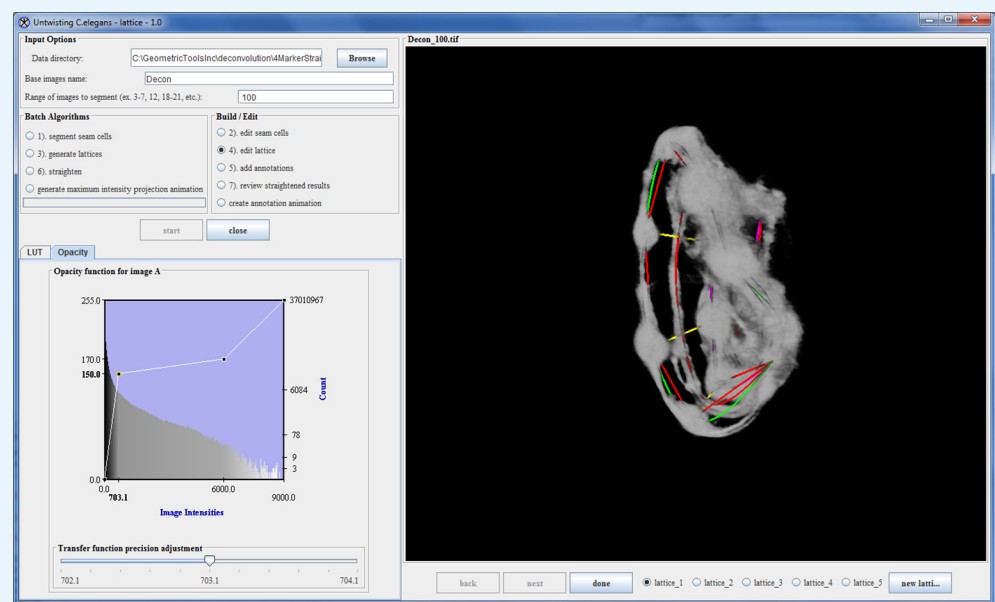

**Appendix 1—figure 5.** Automatic lattice-building output. The correct lattice is listed first as it had the highest rank.

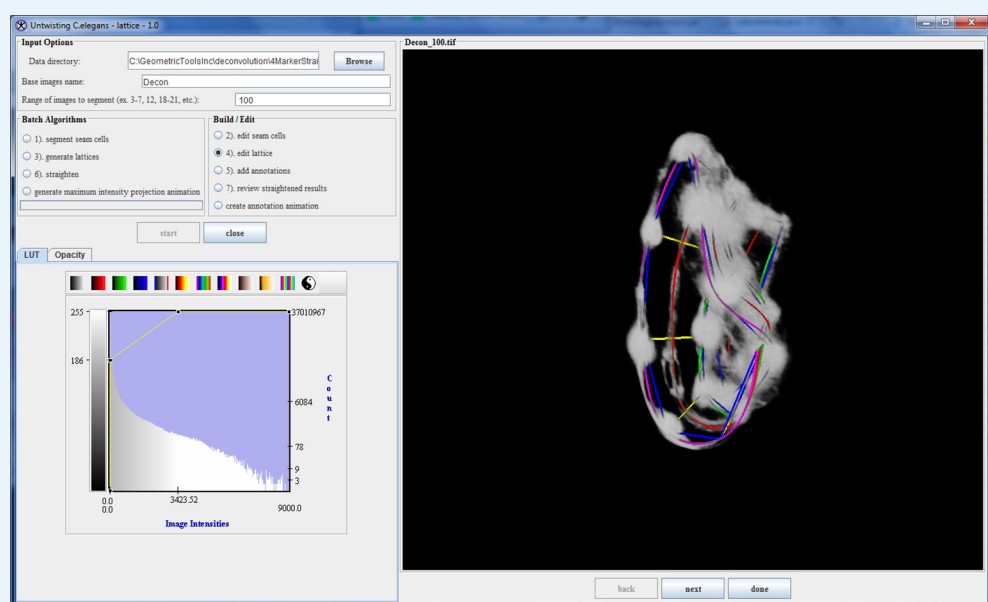

**Appendix 1—figure 6.** The lattice after editing by the user.

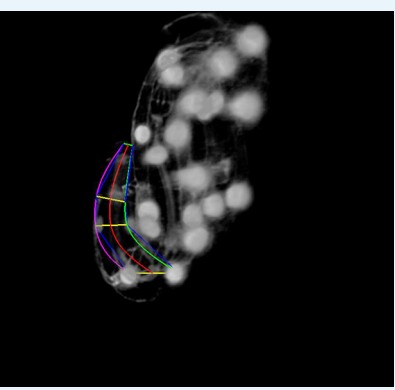

**Appendix 1—figure 7.** The user has started building the lattice starting at the head of the worm.

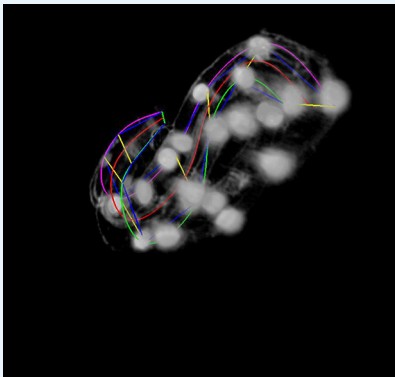

**Appendix 1—figure 8.** More points are added to the lattice. The user rotates the volume to get a better view during this phase. The magenta, red, and green curves represent the left-hand curve, center-line curve, and right-hand curves respectively. The curves are natural splines which are guaranteed to pass through the user-selected lattice points while minimizing bending to produce a smooth curve that matches the shape of the worm fairly accurately.

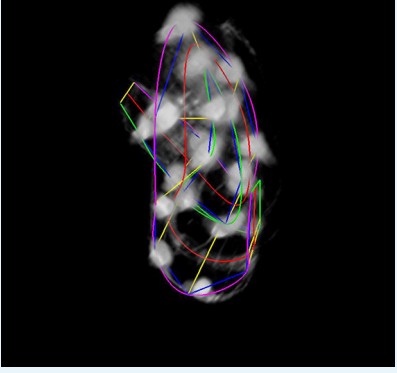

**Appendix 1—figure 9.** The final lattice. Each of the 10 seam-cell pairs is marked, and 8 additional pairs have been added to capture the curve of the worm.

## Interactively placing annotations

The program also features the ability to add annotation points, for which the program records the X, Y, and Z-coordinate in relation to a user-defined origin point (typically the nose), and then outputs that location information in a spreadsheet file. The annotation points can be used to define the position of a cell or cellular structure in each worm volume so spatial displacement can be measured over time. Thus a user can examine the position of a specific cell or structure of interest in multiple worm volumes, to determine how stereotyped the position of that cell is during embryo development (*Appendix 1—figure 10*).

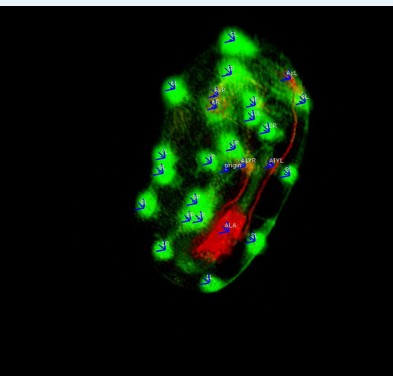

**Appendix 1—figure 10.** Annotations added to the worm volume labeling parts of the neuron.

An integrated rendering capability, which graphically displays the position of the annotation points over time, enables a user to directly visualize how the various annotation points relate to each other in the developing embryo ( *Appendix 1—figure 11*). This capability allows a user to identify the position of a neuron or cell of interest throughout development, and display how that position relates to the rest of the worm and changes over time.

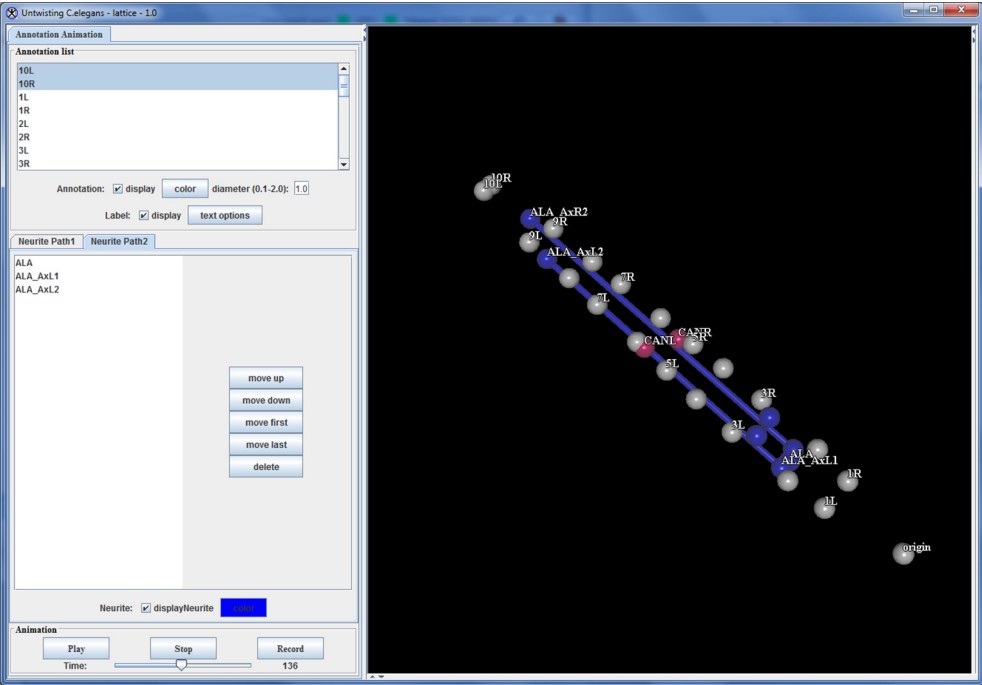

**Appendix 1—figure 11.** Annotation visualization tool displays changes in positions over time.

## Automated model-building

Once the lattice for a worm volume is created, the next step in untwisting the worm volume is to generate a 3D model of the worm from the lattice. The goal is to capture as accurately as possible the 3D shape of the worm. For places where the worm folds back on top of itself and the outer surfaces of the worm touch this is a difficult challenge due to the transparent nature of the worm and lack of a clear boundary between the inside and outside of the worm. This section describes building the worm model and algorithms that attempt to account for overlapping sections of the worm.

Modeling the worm is done automatically, but the user has the option to view the model and interactively modify the lattice to improve the model's accuracy.

To build the worm model, the algorithm interpolates between the lattice points, creating two smooth curves from head to tail along the left and right-hand sides of the worm body. A third curve down the center-line of the worm body is also generated. Eventually, the center-line curve will be used to determine the number of sample points along the length of the straightened worm, and therefore the final length of the straightened worm volume (*Appendix 1—figure 12*).

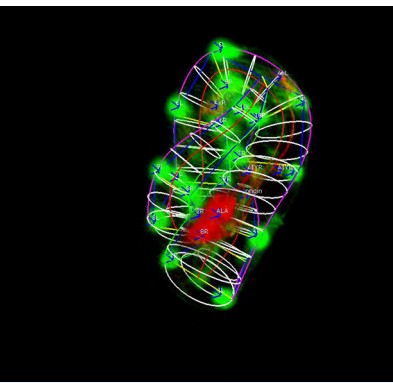

**Appendix 1—figure 12.** The initial ellipse-based model of the worm. The ellipses fit within the boundaries of the natural spline curves.

Given the center-line curve and left and right-side curves, a series of planes is defined. The center-line curve is uniformly sampled with a spacing of one voxel. This point serves as the center of the plane, while the first derivative to the curve serves as the plane normal. The plane horizontal axis is the vector between the corresponding points on the left and right-hand curves. The plane vertical axis is constrained to the axis perpendicular to both the plane normal and horizontal axis. This way the three curves fully define the sample plane in 3D.

Once 2D sampling planes have been determined, each sampling plane is constrained by an elliptical model of the worm cross-section with limits defined by the left- and right-side curves. The elliptical model is tested for regions of the worm where the ellipses overlap and all voxels that fall into overlapping areas are removed, producing a set of new contours. The new contours are then expanded outward from the center, until they either contact an expanding contour line from another region of the worm or reach the limit of the sample plane. This process defines the 2D sampling planes and contours within the planes that are used to create the worm model. *Appendix 1—figure 12* shows the original elliptical model of the worm, *Appendix 1—figure 13* shows the corrected and expanded contours, and *Appendix 1—figures 14–16* show a solid version of the worm model.

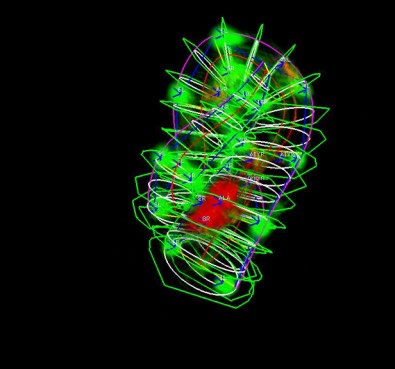

**Appendix 1—figure 13.** The expanded worm model. The original ellipses are expanded until they contact an adjacent surface of the worm or they reach the boundary of the sample plane.

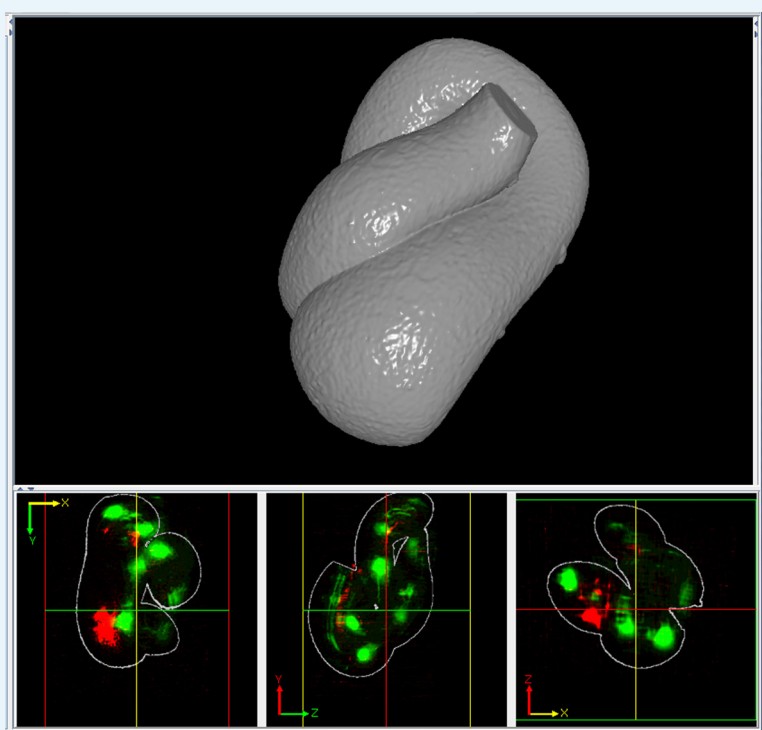

**Appendix 1—figure 14.** A solid representation of the worm surface. The outlines in the bottom three panels show how the surface encapsulates the volume data.

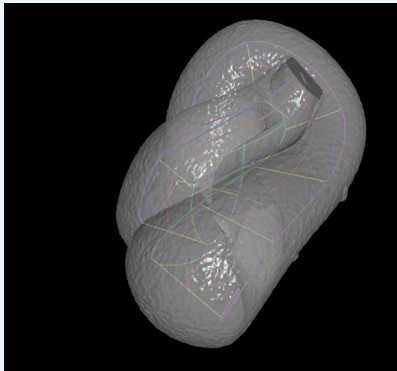

**Appendix 1—figure 15.** A semi-transparent view of the worm surface model, with the lattice shown inside.

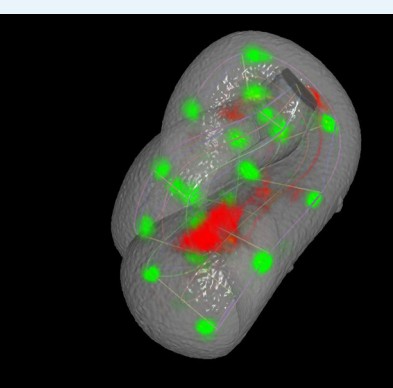

**Appendix 1—figure 16.** A semi-transparent view of the worm surface model displaying lattice curves, fluorescently- labeled seam cell nuclei, and neuron.

During the expansion process a segmented model of the left-right fluorescent markers is used to improve accuracy (*Appendix 1—figure 17*). The segmented marker image labels all voxels that fall inside the fluorescent markers with a unique marker ID. This helps distinguish adjacent edges of the worm where markers may come into contact, prevents one edge of the model from expanding and capturing data from a contacting surface instead of stopping at the correct surface boundary, and ensures that the entire segmented marker is included in the worm model. The image-based corrections require that data from all color channels be processed simultaneously.

## Further details in a step-by-step format

1. All lattices as well as any annotation statistics are saved to file for future analysis.

2. The fluorescent markers in the worm volume are automatically labeled with a corresponding marker ID. (*Appendix 1—figure 17*)

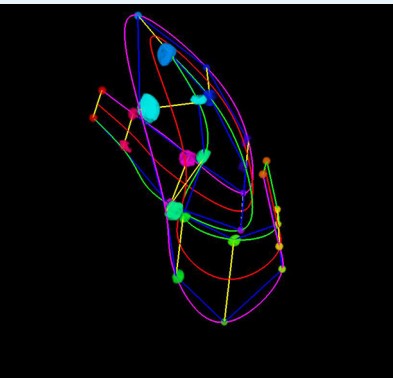

**Appendix 1—figure 17.** Labeled fluorescent markers. Each pair has a unique color value, indicating which pairs belong on the same slice in the final straightened image. Labeling the lattice pairs this way helps disambiguate voxels with potential conflicts.

3. The center line of the worm is calculated from the midpoint between the left and right points of the lattice.

4. Three curves are generated from the three sets of points (left, center, right) using natural splines to fit the points. Natural splines generate curves that pass through the control points, have continuous first and second derivatives and minimize the bending between points (*Appendix 1—figure 9*).

5. The center curve is uniformly sampled along the length of the curve. The step size is set to be one voxel. This determines the length of the final straightened image and ensures that each slice in the straightened image is equally spaced. The points along the curve are the center-points of the output slices.

6. Each spline can be parameterized with the parameter $t$, where the start of the curve has $t$ = 0 and the end of the curve has $t$ = 1. $t$ is calculated on the center curve, and used to determine the corresponding locations on the left and right hand curves, which may be longer or shorter than the center curve, depending on how the worm bends. Using the parameterization ensures that the left and right hand curves are sampled the same number of times as the center curve and that the points from start to end on all curves are included.

7. Given the current point on the center curve and the corresponding positions on the left and right hand curves, the 2D sampling plane can be defined. The center point of the plane is the current point on the center curve. The plane normal is the first derivative of the center line spline. The plane horizontal axis is the vector from the position on the left hand curve to the position on the right hand curve. The plane vertical axis is the cross-product of the plane normal with the plane horizontal axis. This method fully defines the sample plane location and orientation as it sweeps through the 3D volume of the worm.

8. Once the sample planes are defined, the worm cross-section within each plane needs to be determined. Without a model of the worm cross-section the sample planes will overlap in areas where the worm folds back on itself. The first step in modeling the worm cross-section is to define an ellipse within each sample plane, centered in the plane. The long axis of the ellipse is parallel to the horizontal axis of the sample plane. The length is the distance between the left and right hand points. The ellipse short axis is in the direction of the plane vertical axis; the length is set to 1/2 the length of the ellipse long axis. This ellipse-based model approximates the overall shape of the worm, however it cannot model how the worm shape changes where sections of the worm press against each other. The next step of the algorithm attempts to solve this problem. (*Appendix 1—figure 12*).

9. The set of ellipses from the head of the worm to the tail defines an approximate outer boundary of the worm in 3D. The centers of each ellipse are spaced one voxel apart along the center line curve of the worm, and each ellipse corresponds to a single output slice in the final straightened image. This step generates a model of the worm where each voxel that falls within one of the ellipses is labeled with the corresponding output slice value. Voxels where multiple ellipses intersect are labeled as conflict voxels. Once all ellipses have been evaluated, the conflict voxels are removed from the model.

10. The marker segmentation image is used to resolve conflicts where multiple ellipses overlap. Each slice in the output image should extend only to the edges of the left-right markers for the corresponding region of the worm volume. This prevents a slice from extending beyond the worm boundary and capturing the adjacent worm region. Because the marker segmentation image only segments the left-right markers, it is not possible to resolve all possible conflicts.

11. The last step is an attempt to ensure that as much of the worm data is captured by the algorithm as possible. Using the marker segmentation image where possible as a guide to the worm boundary, each slice of worm model is grown outward. The points on the boundary are expanded in an iterative process until the point comes in contact with another edge of the worm. For areas of the worm where it folds back on itself, this process results in a flattened cross-section where the folds press against each other, matching images observed in electron microscopy data (*Appendix 1—figure 18*). For areas of the worm where the cross-section does not contact other sections of the worm, the 2D contour extends outward until it reaches the edge of the sample plane, capturing as much data as possible.

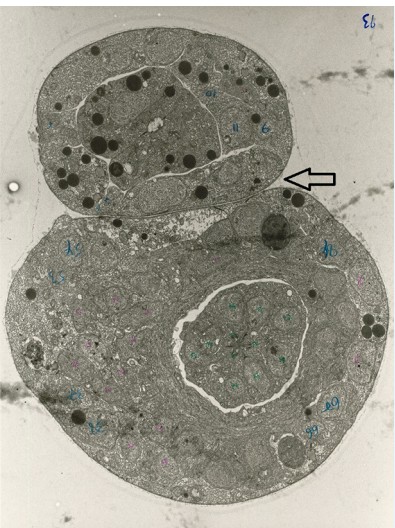

**Appendix 1—figure 18.** An EM image of the worm shows the worm body is flattened where overlapping segments come into contact.

## Untwisting the worm

Once the 3D model of the worm is finalized, a 2D slice plane is swept through the model. At each sample-point along the 3D center spline of the worm, the 2D plane is intersected with the original 3D volume. Voxels that fall outside the updated 2D worm contour are set to the image minimum value (typically = 0). Voxels that fall inside the 2D worm contour are copied into the output slice. The set of 2D slices from the worm head to tail are concatenated to form the final 3D straightened volume.

During the straightening step, as well as during the model-building process or when the 2D sample plane is intersected with the 3D volume, steps are taken by the algorithm to minimize sampling artifacts. Due to the twisted configuration of the worm, sampling the volume along the outer-edge of a curve will cause under-sampling of the data while the inside edge of the curve will be over-sampled.

To reduce sampling artifacts, the sample planes are interpolated between sample points, using the maximum distance between points along consecutive contours to determine the amount of super-sampling. The multiple sample planes are averaged to produce the final slice in the straightened image. In addition, each contour is modeled as having the thickness of one voxel and sample points that fall between voxels in the volume are tri-linearly interpolated.

