## [Decision Letter]

Thank you for submitting your work entitled "Untwisting the *Caenorhabditis elegans* embryo" for peer review at *eLife*. Your submission has been favorably evaluated by K VijayRaghavan (Senior editor), a Reviewing editor, and two reviewers.

The reviewers have discussed the reviews with one another and the Reviewing editor has drafted this decision to help you prepare a revised submission.

Summary:

The manuscript by Christensen et al. describes important technical advances in image analysis that will be important in delineating late embryonic developmental events in *C. elegans*, especially nervous system development. Previous attempts to analyze these stages (in which many key processes of differentiation and circuit formation occur) have been obscured by factors such as muscle movements and the unpredictably folded conformation of the embryo as it elongates. Christensen et al. show that advances in imaging and image analysis can overcome these obstacles, allowing them to generate an 'untwisted' straightened view of the late embryo over time. Although this is laborious (15 h work per embryo) it is an important proof of principle and will open up this phase of development to analysis. The manuscript is thorough and well explained. This work will be of interest to the *C. elegans* and 'developmental imaging' communities.

Requested revisions (in no order of importance; most of them quite minor):

General points:

1) The manuscript appears quite wordy and disproportionate to the lesson that the reader should extract. Please try to provide more concise descriptions.

2) The untwisted volumetric images of multiple animals need to be compared.

Specific points:

1) Nature of the folding/twisting of the embryo: This may be a minor semantic point but are worm embryos truly twisted (helically, around their axis) or folded/bent, or both? In larval development the body can only flex in the dorsoventral axis, it would be interesting if the embryo does undergo some helical twisting. The authors now have the first accurate dataset on this and it is an omission that ether is not a clearer account of how embryos twist or flex (just a rough statement that it is 'more complex' than in larvae.

2) Seam cell marker is introduced correctly as a marker for seam cell nuclei but later on they are just 'seam cells' which could confuse non-experts. The seam cells are doing a lot that is not captured by the nuclei positions.

3) How many points must be in the lattice for this to work? Can you define a minimum number or does accuracy increase with number of lattice points?

4) Seam cell V5 (Results, sixth paragraph) is presumably Q/V5, which divides in the elongating embryo (Sulston, 1983).

5) Elongation (as displacement of seam cell nuclei) is nicely shown to be biphasic. There have been relevant analyses in some papers on alpha spectrin, e.g. Norman and Moerman, that could be cited.

6) Results, eleventh paragraph: The nose is the anterior terminus of the pharynx (well, buccal cavity) not the intestine.

7) Seam cell displacement is described in the same paragraph of the Results as being 'at variable speeds' but I think the authors mean that different seam cell nuclei separate at a variety of speeds. Although from the data mentioned in the text it is not clear if the speed is different or the duration of movement is different.

8) The authors make a key point that elongation is non uniform along the axis, in that head seam cells separate more than body seam cells. This may not have been explicitly stated before elsewhere but is a natural consequence of the morphology of the comma stage embryo, where the head is thicker than the body so must elongate more. Thus elongation is mostly of the head or H cells, in contrast to postembryonic growth which is mostly in the body. Perhaps more could be made of these data and whether this relates to H vs. V cells seam fates.

9) The authors could discuss whether the approaches developed here could be applied to any other developmental systems.

10) A link to the open-source code, compiled software, and a tutorial to use both, must be present at the start of the paper, perhaps even in the Abstract (as well as other places within the paper). Since websites can go offline for a variety of reasons, it would be best practice to make the code, software, and manuals available as supplements hosted at *eLife* or perhaps on the Galaxy platform.

11) The authors perform some segmentation of seam cells and the Bao lab has produced software that tracks, in semi-automated fashion, embryonic cell divisions. The authors should address why they did not implement such a method here. In other words, why must every image volume be manually annotated such that a developing embryo requires ~15 hours of manual annotation? Is it not possible to annotate the first volume, then automatically track the seam cells in subsequent volumes thereafter (requiring manual input only when necessary)?

12) Figure 1, left side: The green spline is hard to see above the green fluorescent markers, it would be best to choose another color.

13) Figure 1—figure supplement 2) refers to a larva, not an "embryo". Also, it would be best practice to identify the larval stage.

14) Results, fifth paragraph: Measurement differences between twisted and untwisted seam cell pairs and the pharyngeal contour need to be quantified as variance or SEM across the 5 profiled embryos as opposed to stating "typical" values. The authors can include mean, max, median, mode… as well if they feel these give an additional perspective on the accuracy of their untwisting method. The same is true for the ninth paragraph and Figure 4—figure supplement 3, wherein we are given a qualitative term "most" instead of a quantifiable statistic. Some of this data is already present in [Supplementary-material SD4-data] and [Supplementary-material SD5-data] and should be used in place of the qualitative statements. This is present again in the eleventh paragraph where the tilde symbol is used instead of quantifying the mean and some form of variance (SD or SEM). And, once again, in the Methods, last paragraph.

15) Figure 22 claims that each segmented fluorescent marker pair has a unique color value but that is neither apparent, nor visible in the accompanying picture. Furthermore, algorithm detail #8 states that the short axis of the ellipses comprising the embryonic boundary is half the distance of long axis (between seam cell pairs). As shown in their accompanying EM image (Figure 24), the embryonic cross section is more rounded than this and so the estimated value appears much smaller than necessary. The authors should address this discrepancy. There is a possibility that this choice is affecting the X and Y measurements in the main Figure 3, the authors should address whether the choice of the ellipse's short-axis estimate changes these values.

16) Figure 3 is a phenomenal figure, capturing the power of the authors’ technique. If possible though, it would be helpful to have, in addition to the embryonic stage, the time since the first embryonic division superimposed at the corners of these images (E–H).

17) Results, eighth paragraph: It would be helpful if the authors could discuss potential causes for the time-shifts in elongation course observed between different embryos. Were developmental conditions strictly controlled or is this perhaps a consequence of variables such as maternal effects (starvation, contamination, plate age,…), room temperature, heat from the diSPIM, etc.? This would also help to elucidate what may have caused embryo 5 to have such a radically different migration for the CANs.

18) Results, tenth paragraph: To give proper perspective on these fits, it would be best if the authors could provide the dimensions of the egg. This should be easily recoverable from their image volumes.

19) Figure 5: I believe the axial distance panel annotations are reversed as it appears that panel B represents the least distance and D the greatest.

20) Methods, Strains: The promoters should be identified for the SCM and Coelomocyte markers. Future papers may find that these promoters contribute to deviations from wild-type development. Moreover, for the remaining promoters, the authors should provide forward/reverse primers, genomic locations, and/or paper citations wherein this information can be found.

21) Methods, Data Acquisition: The MIPAV site has been updated and the old link appears to be dead.

---

## [Author Response]

*General points:*

*1) The manuscript appears quite wordy and disproportionate to the lesson that the reader should extract. Please try to provide more concise descriptions.*

We have attempted to make the manuscript more concise, while still incorporating new content asked for by the reviewers. Despite this added content we were able to keep the new version of the paper approximately the same length as the old version.

*2) The untwisted volumetric images of multiple animals need to be compared.*

We have added Figure 3—figure supplement 1 and Figure 3—figure supplement 2 to directly compare representative volumes from five 1.5-fold and 3-fold embryos (these new figure supplements are described in the eighth paragraph of the Results). Importantly, the images show that overall morphology and seam cell positions are similar among embryos.

*Specific points:*

*1) Nature of the folding/twisting of the embryo: This may be a minor semantic point but are worm embryos truly twisted (helically, around their axis) or folded/bent, or both? In larval development the body can only flex in the dorsoventral axis, it would be interesting if the embryo does undergo some helical twisting. The authors now have the first accurate dataset on this and it is an omission that ether is not a clearer account of how embryos twist or flex (just a rough statement that it is 'more complex' than in larvae.*

We have closely inspected our data and found evidence for helical twisting. We highlight evidence for this claim in Figure 1—figure supplement 1, and mention it in the main text: “Nematode embryos undergo both bending and helical twisting around the nose-to-tail axis (Figure 1—figure supplement 1) posing challenges in untwisting the embryo relative to larval or adult nematodes”. Although interesting, an in-depth study of helical twist is outside the scope of this work, so we have not further investigated this point in detail.

*2) Seam cell marker is introduced correctly as a marker for seam cell nuclei but later on they are just 'seam cells' which could confuse non-experts. The seam cells are doing a lot that is not captured by the nuclei positions.*

We thank the reviewers for raising this point and have updated the text so that all previously incorrect instances of ‘seam cells’ are referred to as ‘seam cell nuclei’.

*3) How many points must be in the lattice for this to work? Can you define a minimum number or does accuracy increase with number of lattice points?*

We thank the reviewers for raising this point, which we have now clarified in the third paragraph of the Results, and visually in Figure 1—figure supplement 3. The highest-quality lattices are generated when user builds a lattice that closely matches the morphology of the embryo, e.g. as quantified by the sides of the generated worm model having significant overlap with the actual sides of the embryo, the generated midline of the worm model following the actual midline of the embryo, etc. These requirements are best achieved by using as many lattice points as required to get a good match between the model and the actual data; depending on the degree of twisting/bending observed in a volume the number of required points needed to optimally capture the worm morphology may differ. In practice, lattices that match well with actual embryo morphology incorporate lattice points at each seam cell nucleus (for a total of 20-22 points), a pair of points at the nose (2 points), and as many points as required to capture bends occurring between seam cell nuclei (usually 2B, where B is the number of bends between nuclei in the volume). Hypodermal labeling past the tip of the tail is usually visible only in some volumes, but when observed, another pair of points is often placed at the tip of the tail (2 points). Thus, the minimum number of points required to generate a lattice (assuming 20 seam cell nuclei and no pair of tail points) is 22+2B. Using more points (more than 32) may provide better agreement between the generated model and the actual worm, but it usually leads only to minor changes in practice, and in our experience is not time-effective.

*4) Seam cell V5 (Results, sixth paragraph) is presumably Q/V5, which divides in the elongating embryo (Sulston, 1983).*

We thank the reviewers for pointing this out; after division we tracked the anteriormost daughter cell Q, and have clarified this in the eighth paragraph of the Results. We now refer to this cell as Q/V5 throughout the text.

*5) Elongation (as displacement of seam cell nuclei) is nicely shown to be biphasic. There have been relevant analyses in some papers on alpha spectrin, e.g. Norman and Moerman, that could be cited.*

In reference to this point, we have now cited:

a) Norman K.R. and Moerman, D.G. (2002). Alpha spectrin is essential for morphogenesis and body wall muscle formation in *Caenorhabditis elegans*. Journal of Cell Biology 157((O'Donnell et al., 2009)): 665-677.

b) Chin-Sang, I.D., and Chisholm, A.D. (2000). Form of the worm: genetics of epidermal morphogenesis in *C. elegans*. Trends Genetics 16((Holekamp et al., 2008)): 544-551.

c) Ding, M., Woo, W-M., Chisholm, A.D. (2004). The cytoskeleton and epidermal morphogenesis in *C. elegans*. Experimental Cell Research. 301((Kolodkin and Tessier-Lavigne, 2011)): 84-90.

d) Priess, J.R., and Hirsh, D.I. (1986). *Caenorhabditis elegans* morphogenesis: The role of the cytoskeleton in elongation of the embryo. Dev. Biol. 117((Kolodkin and Tessier-Lavigne, 2011)): 156-173.

*6) Results, eleventh paragraph: The nose is the anterior terminus of the pharynx (well, buccal cavity) not the intestine.*

We thank the reviewers for this correction. We have fixed this mistake.

*7) Seam cell displacement is described in the same paragraph of the Results as being 'at variable speeds' but I think the authors mean that different seam cell nuclei separate at a variety of speeds. Although from the data mentioned in the text it is not clear if the speed is different or the duration of movement is different.*

We have clarified what we meant, that over the same time period (elongation), some pairs of adjacent seam cell nuclei separate from each other over significantly greater distances than do other adjacent seam cell nuclear pairs, so that the average speed of separation of nuclei near the head is greater than the average speed of separation of nuclei near body or tail.

*8) The authors make a key point that elongation is non uniform along the axis, in that head seam cells separate more than body seam cells. This may not have been explicitly stated before elsewhere but is a natural consequence of the morphology of the comma stage embryo, where the head is thicker than the body so must elongate more. Thus elongation is mostly of the head or H cells, in contrast to postembryonic growth which is mostly in the body. Perhaps more could be made of these data and whether this relates to H vs. V cells seam fates.*

We now mention in the eleventh paragraph of the Results that the non-uniformity of elongation along the axis is a natural consequence of morphology of the comma stage embryo. We prefer not to speculate about whether this point relates to H vs. V seam cell fates.

*9) The authors could discuss whether the approaches developed here could be applied to any other developmental systems.*

Although the precise details of our algorithm are immediately applicable only to other systems with vermiform geometry, some of the other concepts we implement (alignment and pooling of distinct datasets derived from multiple animals) are likely more general. We have mentioned this point in the fourth paragraph of the Discussion.

*10) A link to the open-source code, compiled software, and a tutorial to use both, must be present at the start of the paper, perhaps even in the Abstract (as well as other places within the paper). Since websites can go offline for a variety of reasons, it would be best practice to make the code, software, and manuals available as supplements hosted at* eLife *or perhaps on the Galaxy platform.*

The untwisting plugin has been implemented as part of MIPAV, which has been a long-running and stable (in use for the last 15 years) graphics processing program. Hosting on the MIPAV site allows basic users to download the plugin without needing to compile the code, allows advanced users access to information about how to look at the code if they are interested, and finally allows us to continue updating and improving the program without needing to change the provided link. Because of these features, we feel that the MIPAV site is best for hosting the plugin, and have made it available at http://mipav.cit.nih.gov/plugin_jws/mipav_worm_plugin.php. We describe the link in the Abstract, Introduction,, Results and Methods. We have also written a tutorial describing how to use the untwisting plugin, which we have included as [Supplementary-material SD1-data] and which we reference in the Abstract and Results.

*11) The authors perform some segmentation of seam cells and the Bao lab has produced software that tracks, in semi-automated fashion, embryonic cell divisions. The authors should address why they did not implement such a method here. In other words, why must every image volume be manually annotated such that a developing embryo requires ~15 hours of manual annotation? Is it not possible to annotate the first volume, then automatically track the seam cells in subsequent volumes thereafter (requiring manual input only when necessary)?*

We have taken this point very seriously, and are happy to report that we have improved the automation of our approach. We now automatically detect and segment seam cells, and automatically generate a list of candidate lattices for each timepoint. The user intervenes at two editing steps, to ensure the accuracy of segmentation and lattice generation. Conservatively, we estimate this improvement results in ~1/2 as much manual user effort as before. Our improved, semi-automated approach is discussed in the Results, and [Supplementary-material SD2-data]. Due to the rapid post-twitching movement, automatically tracking seam cell nuclei in the same manner that histones are tracked for lineaging would require imaging rates on the order of 1 volume every 3-4 seconds, which for many applications implies an unnecessary increase in dose to the embryo, and also greatly increases the amount of data that needs to be recorded and processed.

*12) Figure 1, left side: The green spline is hard to see above the green fluorescent markers, it would be best to choose another color.*

We thank the reviewers for this comment and have drawn over the green spline in a more visible color.

*13) Figure 1—figure supplement 2) refers to a larva, not an "embryo". Also, it would be best practice to identify the larval stage.*

We have clarified this point in the fourth paragraph of the Results and identified the larval stage as L2 explicitly in the Figure 1—figure supplement 4 caption.

*14) Results, fifth paragraph: Measurement differences between twisted and untwisted seam cell pairs and the pharyngeal contour need to be quantified as variance or SEM across the 5 profiled embryos as opposed to stating "typical" values. The authors can include mean, max, median, mode… as well if they feel these give an additional perspective on the accuracy of their untwisting method. The same is true for the ninth paragraph and Figure 4—figure supplement 3, wherein we are given a qualitative term "most" instead of a quantifiable statistic. Some of this data is already present in [Supplementary-material SD4-data] and [Supplementary-material SD5-data] and should be used in place of the qualitative statements. This is present again in the eleventh paragraph where the tilde symbol is used instead of quantifying the mean and some form of variance (SD or SEM). And, once again, in the Methods, last paragraph.*

We have made all such qualitative statements quantitative, by providing appropriate statistics for the requested data. Instances include:

Results, seventh paragraph: We have quantified the population average difference and standard deviation across time and embryo number (<μ_Difference, time_>_embryo_ population standard deviation <σ_Difference, time_>_embryo_) of the twisted vs. untwisted difference for H0, T, and pharyngeal length.

Results, tenth paragraph: We have quantified average standard deviation calculated across all 20 seam cell nuclei and all timepoints, <<σ_X_>_time_>_seam cell_, <<σ_Y_>_time_>_seam cell_, and <<σ_Z_>_time_>_seam cell_.

Results, eleventh paragraph: We have quantified the mean and standard deviation of elongation rate for the non-origin-H0 and V6-T seam cell pairs.

In the subsection “Validating fits for each embryo”: We have quantified the absolute differences between averaged and fitted coordinates at each time point, and then calculated the means and standard deviations of these differences across time, reporting them as μ _avg-fit, time_ and σ _avg-fit, time_ in Supplementary file 6. We then computed the average over all seam cell nuclei of these average differences to generate <μ _Xavg-Xfit, time_>_seam cell_; <μ _Yavg-Yfit, time_>_seam cell_; and <μ _Zavg-Zfit, time_>_seam cell_.

For Figure 4—figure supplement 3, we have quantified in the legend axial position stereotypy across the 20 seam cells via <<σ_Z_>_time_>_seam cell_, and axial stereotypy for CANL via <σ_Z_>_time_.

We have also added a section to the Methods describing our population statistics calculations, (“Population statistics”).

15) Figure 22 claims that each segmented fluorescent marker pair has a unique color value but that is neither apparent, nor visible in the accompanying picture. Furthermore, algorithm detail #8 states that the short axis of the ellipses comprising the embryonic boundary is half the distance of long axis (between seam cell pairs). As shown in their accompanying EM image (Figure 24), the embryonic cross section is more rounded than this and so the estimated value appears much smaller than necessary. The authors should address this discrepancy. There is a possibility that this choice is affecting the X and Y measurements in the main Figure 3, the authors should address whether the choice of the ellipse's short-axis estimate changes these values.

We thank the reviewers for noticing the color value issue and have fixed it in Figure 22. Regarding the elliptical cross-section, the model intentionally starts out smaller than expected, and then is expanded to capture the rest of the worm. This procedure is implemented to deal with cases where parts of the embryo are “squished” against each other, which would otherwise inappropriately capture the abutting surface of the worm. Thus, the initial elliptical cross-section itself is not directly applied to raw data. We have clarified this point in [Supplementary-material SD2-data].

*16) Figure 3 is a phenomenal figure, capturing the power of the authors’ technique. If possible though, it would be helpful to have, in addition to the embryonic stage, the time since the first embryonic division superimposed at the corners of these images (E-H).*

We thank the reviewers for the positive feedback. Embryos were roughly staged based on morphology, but were not tracked from the two-cell stage so precisely defining time since the first embryonic division unfortunately is not practical with the reported datasets. We have therefore left this figure unchanged. However, the reviewer’s point is well taken and for future applications we agree that more rigorous assessment of timing relative to the first embryonic division is a good idea.

*17) Results, eighth paragraph: It would be helpful if the authors could discuss potential causes for the time-shifts in elongation course observed between different embryos. Were developmental conditions strictly controlled or is this perhaps a consequence of variables such as maternal effects (starvation, contamination, plate age,…), room temperature, heat from the diSPIM, etc.? This would also help to elucidate what may have caused embryo 5 to have such a radically different migration for the CANs.*

As the reviewers suggest, the cause of the variability relating to shifting is interesting. Several potential causes suggest themselves. As described in the response to point 16, embryo development was not tracked from the two-cell stage, so it is likely that embryos were at slightly different ages at the onset of imaging. Temperature and other environmental conditions such as plate age were moderately controlled—temperature may have fluctuated by 2-3 degrees C while keeping strains and during imaging—and plates were used on consecutive days so plate ages may vary by up to a day. Finally, intrinsic variables like imaging conditions or health of the mother animal or embryo could have played a role in the variability we observed. We have described these issues in the second paragraph of the Discussion.

*18) Results, tenth paragraph: To give proper perspective on these fits, it would be best if the authors could provide the dimensions of the egg. This should be easily recoverable from their image volumes.*

As the length of the elongating embryo is much greater than the length of the egg, and the diameter is much smaller, we believe that growing embryo dimensions are probably a better metric than egg dimensions for examining fits. We have thus provided the average length and diameter of an untwisted embryo, calculated from the final volumes of the five embryos, in the tenth paragraph of the Results.

*19) Figure 5: I believe the axial distance panel annotations are reversed as it appears that panel B represents the least distance and D the greatest.*

The reviewers are correct that there is an error, but the correction is that panels B and C should be swapped. These panels show the change in distance between measured seam cells from the beginning of elongation to the end; thus graphs with a high rise indicate greatest distance and those with low rise indicate the least. Thus, origin-H0 and H1-H2 show the greatest increase in distance while V6-T shows the least increase in distance. We have adjusted the figure accordingly.

*20) Methods, Strains: The promoters should be identified for the SCM and Coelomocyte markers. Future papers may find that these promoters contribute to deviations from wild-type development. Moreover, for the remaining promoters, the authors should provide forward/reverse primers, genomic locations, and/or paper citations wherein this information can be found.*

More detail is now provided in the subsection “Strains”. Most of the constructs used were ordered from the CGC and represent preexisting reagents/strains created by other labs. We have included primer information for *olaex 2457*, which was created in house, and have provided references for constructs originally created by other labs (Lakadamyali et al., 2012; Totong, Achilleos and Nance, 2007; Terns et al., 1997; Koh and Rothman, 2001). The SCM promoter, to our knowledge, has not been identified more definitively or assigned to a specific gene.

21) Methods, Data Acquisition: The MIPAV site has been updated and the old link appears to be dead.

We have compiled the plugin code and are hosting it on the MIPAV website. The link to it is: http://mipav.cit.nih.gov/plugin_jws/mipav_worm_plugin.php